

# Climate change impacts on hydroclimatic regimes and extremes over Andean basins in central Chile

Deniz Bozkurt[1], Maisa Rojas[1,2], Juan Pablo Boisier[1], and Jonás Valdivieso[3]

[1]Center for Climate and Resilience Research, University of Chile, Santiago, Chile
[2]Department of Geophysics, University of Chile, Santiago, Chile
[3]Department of Civil Engineering, University of Chile, Santiago, Chile

*Correspondence to:* Deniz Bozkurt (dbozkurt@dgf.uchile.cl)

**Abstract.** This study examines the projections of hydroclimatic regimes and extremes over Andean basins in central Chile ($\sim 30 - 40^o$S). We have used daily precipitation and temperature data based on observations to drive and validate the VIC macro-scale hydrological model in the region of interest at a $0.25^o$ x $0.25^o$ resolution. Historical (1960-2005) and projected, following the RCP8.5 scenario (2006-2099), daily precipitation and temperatures from 26 CMIP5 climate models are bias

corrected and used to drive the VIC model to obtain regional hydroclimate projections. Following the robust drying and warming shown by CMIP5 models in this region, the VIC model simulations indicate decreases in annual runoff of about 40% by the end of the century, larger that the projected precipitation decreases (up to 30%). Center timing of runoff shifts to earlier dates, 3-5 weeks by the end of the century. The Andes snowpack is projected to be less than half of the reference period by mid-century. The projected hydroclimatic regime is also expected to increase the severity and frequency of extreme events.

The probability of having extended droughts, such as the recently experienced mega-drought (2010-2015), increases to up to 5 events/100 years. On the other hand, probability density function of annual maximum daily runoff indicates an increase in the frequency of flood events. The estimated return periods of annual maximum runoff events depict more drastic changes and increase in the flood risk as longer return periods are considered (e.g. 25-yr and 50-yr).

## 1 Introduction

Central Chile (hereafter CC), a narrow strip of land between the subtropical southeast Pacific and the Andes cordillera ($30^o$-$38^o$S, Fig. 1), is the heartland of Chile with most of the country's population (around 15 million) and economic activities. The region harbors more than 75% of the country's total irrigated agriculture and the majority of the water reservoirs (Demaria et al., 2013b). Moreover, the winter Andes snowpack within the region provides a wide variety of socioeconomic and environmental benefits, notably the freshwater supply for human consumption, irrigation and hydropower production (Masiokas et al., 2006).

CC features a semi-arid Mediterranean climate and its inhabitants are very much dependent on the 200-1000 mm or so of annual rainfall (Ruttlant and Fuenzalida, 1991). The region has been facing a dramatic rainfall decline during the last decades and a persistent drought has recently (2010-2015) parched CC with a precipitation deficit of approximately 30% (CR2, 2015; Boisier et al., 2016). As temperature-related effects are expected to yield changes in snow cover and seasonality of runoff over snowmelt dominated basins (e.g., Stewart et al., 2005; Adam et al., 2009; Sen et al., 2011), observed temperature trend also





plays a key role in determining regional hydroclimate variability of CC. Indeed, Falvey and Garreaud (2009) highlighted that there is a clear warming trend in the Andes cordillera of CC for the period 1979-2006. Furthermore, Cortés et al. (2011) stated that a negative center time (CT) trend (CT shifting towards earlier dates) was detected for rivers south of $33^o$S in the western Andes. On the other hand, the same study (Cortés et al., 2011) highlighted that there has been no significant change in CT for

high-elevated basins (between $30^o$S and $33^o$S), despite documented warming trends at high elevation sites.

Human-induced future climate change is of particular concern in CC due to both the detected hydroclimatic trends in the region and the crucial importance of water resources availability for drinking water, irrigation and energy. Furthermore, the region is classified as a Mediterranean-type climate in the world and has the potential to be greatly influenced by climate change impacts, similar to California, western Australia, southern France, Spain, Greece and Turkey (e.g., Giorgi and Lionello,

2008; Hannah et al., 2013; Onol et al., 2014). Indeed, the last report of the Intergovernmental Panel on Climate Change (IPCC, 2014) based on the results of the ensemble Global Circulation Models (GCMs) indicate that the observed decreasing trend in precipitation and increasing trend in surface air temperature over the central Andes of Chile are projected to continue into the twenty-first century.

Although GCMs are commonly used in large-scale climate change impacts studies and their data are easily accessible

for a wide variety of usage, they provide limited information about climate change impacts on hydroclimatology and water resources at basin scales, especially over complex terrains such as CC (e.g., Hansen et al., 2006; Sharma et al., 2007). Therefore, investigating the impacts of projected surface air temperature increase and precipitation decrease on hydroclimatic regimes over Andean basins in CC, requires the use of high-spatial resolution climate information as well as the application of hydrological models (e.g., Vicuña et al., 2011; Demaria et al., 2013b). One common method to overcome the difficulties arising from the

usage of coarse resolution GCMs, is to downscale the GCM fields to high-resolution grids by using statistical bias correction techniques based on a gridded observational dataset (e.g., Ines and Hansen, 2006; Piani et al., 2010).

An observation-based gridded $(0.25^o$x$0.25^o)$ dataset of precipitation and temperature (1948-2008) was recently produced for four basins in CC namely (from north to south) Rapel, Mataquito, Maule and Itata basins (Demaria et al., 2013a). To evaluate their dataset, Demaria et al. (2013a) used the Variable Infiltration Capacity hydrological model (VIC) to contrast simulated

streamflows against observations. They concluded that the gridded observations-based dataset can be successfully used for hydrologic simulations of climate variability and change in CC. Demaria et al. (2013b) also used the validated VIC model to assess the climate change impacts on the hydrology of the Mataquito basin in CC based on 12 CMIP3 and CMIP5 GCM outputs. They highlighted that a drier and warmer future will shift the snow line upward (from 2000 m to 2700 m) and reduce the number of days with precipitation.

The present study follows Demaria et al. (2013b) and extends the hydrological assessment to four basins in CC: Rapel, Mataquito, Maule and Itata with the use of more than 20 GCM outputs. This extension allows for a comparison of changes in hydroclimatic regimes and extremes for basins with different characteristics (e.g., snowmelt-dominated and rainfed). Our approach uses also the observations-based gridded daily precipitation and temperature of Demaria et al. (2013a) as a reference dataset to downscale 26 GCM outputs from the Coupled Model Intercomparison Project phase 5 (CMIP5) (Taylor et al., 2012)

via the bias correction technique described in Piani et al. (2010). Adjusted historical (1960-2005) and projected (RCP8.5,



2006-2099) daily precipitation and temperatures from 26 CMIP5 models are used to drive the validated VIC model in order to present hydroclimate change projections in the region of interest at a $0.25^o$x$0.25^o$ resolution.

In Section 2, we describe the study area, data and the hydrological model used in the study. Section 3 provides the results of runoff validation and projected hydroclimate changes. A summary and conclusions are given in Section 4.

## 2  Study area, data and hydrologic model

### 2.1  Study area

The region of interest is located on the western slope of the Andes Cordillera in CC (33-38$^o$S, Fig. 1) and consists of four basins namely (from north to south) Rapel, Mataquito, Maule and Itata basins. The basins are characterized by a sharp increase in west-east topographic gradient varying between sea level and up to 4000 m for Rapel basin, and 2500-3000 m for the other basins. The Maule basin has the largest river system within the four basins, with a drainage area covering approximately 21100 km$^2$. Rapel, Mataquito and Itata basins drain an area of approximately 13700 km$^2$, 6300 km$^2$ and 11500 km$^2$, respectively. These four basins have a key role in Chile's socioeconomic activities providing water for irrigation and for some of Chile's largest cities. Rapel and Maule watersheed feed also one of the largest hydropower plants in CC.

CC is characterized by a semiarid Mediterranean climate and a contrasted landscape. A marked increasing precipitation gradient from north to south follows the relative influence of the South Pacific Subtropical dry regime and the mid-latitude frontal systems (e.g., Falvey and Garreaud, 2009, 2007). Most of the annual precipitation accumulates during austral winter (JJA), when the mid-latitude frontal systems reach CC. A large interannual precipitation variability in the region is largely controlled by El Niño-Southern Oscillation (ENSO) (Ruttlant and Fuenzalida, 1991; Falvey and Garreaud, 2009). Total annual precipitation is about 300-500 mm, increasing to up to 1500 mm in the southern part of CC and in the Andes foothills (Fig. 2a). The seasonal mean rainfall derived from rain-gauge data highlights the annual cycle in each basin with very little summer precipitation (Fig. 2b). Fig. 2c shows the basin averaged time-series of annual precipitation, from which a persistent deficit in the recent decade is apparent. The period 2010-2015 has been termed the mega-drought because of its spatial and temporal extension (CR2, 2015; Boisier et al., 2016). In terms of snow extension, MODIS/TERRA snow cover (Hall et al., 2006) areal fraction of each basin indicates that the Maule basin has the highest snow coverage fraction ($\sim 31\%$) while the Itata basin is more rainfall-dominated basin with the lowest snow coverage fraction ($\sim 18\%$) during JJA (see Fig. S2a). Whereas the Rapel basin, has the highest amount of snow water equivalent among the four studied basins (see Fig. S5) and it is characterized as a snowmelt-dominated basin.

### 2.2  Data

A recent gridded dataset based on daily observed precipitation, maximum and minimum temperatures at a $0.25^o$ grids (Demaria et al., 2013a) was used to drive and validate the VIC model over the region of interest. Demaria et al. (2013a) interpolated NCEP/NCAR Reanalysis to a finer spatial scale of $0.25^o$ degree based on a product of the Tropical Rainfall Measuring Mission





(TRMM 3B42RT) (Huffman and Coauthors, 2007) for the period 1948-2008. Precipitation gauges were used to correct the inaccuracies in the representation of orographic precipitation distribution. Demaria et al. (2013a) reports that the adjusted daily gridded precipitation dataset realistically represents precipitation distribution in the region and can be successfully used for hydrologic simulations of climate change. For temperature, they used a temperature lapse rate of -6.5$^{o}$C km$^{-1}$ based on the elevation differences between the large reanalysis spatial scale and the elevation in each 0.25$^{o}$ degree grid. A more detailed description of the development of the gridded dataset can be found in the same paper and in Sheffield et al. (2006). In addition to the gridded observations-based dataset, we use rain-gauge and runoff data from the Chilean National Weather Service (Dirección Meteorológia de Chile) and Chilean Water Agency (Dirección General de Aguas (DGA), see Fig. 1 and Fig. 2 for distributions of the runoff and precipitation data, respectively). A quality control and gap-filling procedures were applied to monthly station data (see Boisier et al. (2016)).

Climate change scenarios for daily precipitation, maximum and minimum temperatures were obtained from the CMIP5 (Taylor et al., 2012). Historical (1960-2005) and RCP8.5 runs (the highest emission scenario (Moss and Coauthors, 2010)) from 26 GCMs (see Table S1) were adjusted and used to force the VIC model (see section 2.4).

### 2.3 Methodology

The use of a large ensemble of climate simulations has the advantage of providing a more robust information and an uncertainty range for the chosen emission scenario. However, because of the coarse spatial resolution of the GCMs, we downscale the climate variables first. This step is particularly relevant in CC because of its complex topography. Hence, daily precipitation was downscaled to the 0.25$^{o}$ degree by using the statistical bias correction technique based on Piani et al. (2010). This method imposes the observed statistical intensity distribution onto the modeled climate variables. We did not assume that both the observations and simulated precipitation follow a gamma distribution, but we simply found the empirical relationship. As a first step each model was interpolated onto the 0.25$^{o}$ degree grid used in the observations-based dataset. Then, at each grid point an empirical transfer function is constructed by fitting the curve of the sorted observed versus sorted modeled daily distribution of precipitation.

Figure 3 shows the mean (1976-2005) annual precipitation derived from the observation-based dataset (Fig. 3a), from CMIP5 models (Fig. 3b) and the result from the bias correction (Fig. 3c). Both the mean and the spatial pattern of precipitation at 0.25$^{o}$ grids are captured with the bias correction. The CMIP5 model mean climatology captures to some degree the north to south precipitation gradient, but completely misses the important east-west gradient, and therefore greatly underestimates the precipitation over the Andes. Fig. 3d shows the result of this method for the whole region. With this method, the main biases in the modeled precipitation are corrected, such as the underestimation of dry days and extreme rainfall, overestimation of drizzle and biases in the mean. We tested the procedure by obtaining the transfer function for the 1960-1980 period (calibration period), corrected with the transfer function the 1986-2005 (validation period) and compared those to observations. Satisfactory results of the validation procedure are apparent in all the aspects of the above-mentioned modeled precipitation. The same transfer functions were then used to bias correct the precipitation over the 21$^{st}$ century.





For temperature we used a different approach. Because we have to correct both maximum and minimum temperatures, the transfer function implemented for precipitation cannot be used, because there is no way that the condition of having the minimum temperature lower that the maximum temperature can be enforced. For each grid point and month the climatological mean of the observations and the CMIP5 ensemble mean is calculated, and models are corrected by this factor. Therefore we are only correcting the mean, but not any other aspect of the distribution. Fig. S1a, b show the distribution of the daily temperatures, comparing observations, and models. In particular the simulation of extreme events such as days with temperatures above $30^oC$ of temperature below $0^oC$ are partially improved. For example, considering only the summer temperatures, the CMIP5 overestimate days with temperatures above $30^oC$ (12% compared with 5% in the models), with the correction this index decreases to 8% (not shown).

## 2.4 Hydrologic model description and setup

We employed VIC model, a spatially distributed and process-based hydrologic model (Liang et al., 1994), to assess the hydrological impacts of projected climate change in the four study basins. Both surface energy and water balances are resolved over a grid mesh with varying topography, soil properties and vegetation coverage. Sub-grid heterogeneity (e.g., elevation, land cover) is handled via statistical distributions. The model inputs include land-cover distribution, soil information and meteorological forcing data. The model can be run at global or regional scale by providing a wide variety of data sources such as global or regional gridded meteorological dataset as well as GCM/RCM projections. Typically, the model runs at daily or sub-daily time step with a spatial resolution ranging from $1/24^o$ to $2^o$. The model simulates key hydrologic processes and variables such as evapotranspiration, snow water equivalent, soil moisture and surface and subsurface runoff (baseflow). Runoff is computed through variable infiltration curve. Each of the individual cell runoff and baseflow are collected and routed separately by an offline routing scheme (Lohmann et al., 1998). Parameterization and calibration within VIC is performed primarily through adjustments of infiltration parameters, soil layer thickness and baseflow parameters. In recent years, the VIC model has been widely used in climate change and hydroclimate variability studies at different spatial scales (e.g., Christensen et al., 2004; Shukla and Wood, 2008; Maurer et al., 2009). Detailed information about the VIC model can be found in Liang et al. (1994, 1996).

In this study, the VIC model was applied at daily time step and a spatial resolution of $0.25^o$. Vegetation and soil hydraulic properties are based on values from Sheffield et al. (2006). Infiltration parameters, soil layer thickness and baseflow parameters are based on values from Demaria et al. (2013a), which calibrated and validated the model in the study area. Gridded daily precipitation, maximum and minimum temperature datasets of Demaria et al. (2013a) were used to drive the VIC model (VIC-OBS, see Table 1). As the region of interest does not include large-scale basins, offline routing scheme was not applied in this study, as it would not affect the bulk results. Therefore, we sum daily surface runoff and baseflow from each grid cell together to obtain total runoff (e.g., Shukla and Wood, 2008). Shukla and Wood (2008) also noted that the summed runoff field differs from streamflow only at large spatial scales. Basin scale and temporarily averaged runoff fields are then used for further analysis. As the detailed analysis of model validation against streamflow gauges in the basins was given by Demaria et al. (2013a), we present runoff, snow cover and evapotranspiration comparisons in this study. The simulated runoff outputs were validated



against available observed runoff data in Rapel, Mataquito, Maule and Itata basins. Note that gauges located in Mataquito, Maule and Itata basins represent the whole catchment outlet. Due to lack of stream gauge data at the outlet point of Rapel basin, one of the sub-catchment outlet gauges is used for this basin. Monthly discharge data obtained from the stream gauges are converted to an equivalent monthly unit runoff from the catchment area (discharge multiplied by the time unit conversion

and divided by the total catchment area). Snow cover validation is based on monthly MODIS/TERRA snow cover data with a spatial resolution of $0.05^o$ (Hall et al., 2006). Furthermore, given the difficulty of obtaining accurate evapotranspiration measurement, we use GLEAM v3.0 (Martens et al., 2016) evapotranspiration product as a reference to evaluate the simulated evapotranspiration. After the validation, the hydrologic model was then forced by adjusted daily precipitation, maximum and minimum temperature time series from 26 GCMs for both historical (1960-2005, VIC-CMIP5-HIST) and projected (2006-

2099, VIC-CMIP5-RCP8.5) periods in order to carry out climate change projections. All simulations performed in this study are listed in Table 1.

## 3  Results

Based on a detailed streamflow validation, Demaria et al. (2013a) illustrated a reasonable agreement between the VIC model simulations and observed fields. Therefore, we only show the interannual variability and mean annual cycle of observed runoff

together with VIC-OBS and VIC-CMIP5-HIST. We include also snow cover and evapotranspiration evaluations in the supplementary materials (Fig. S2 and Fig. S3). Then we present the results of hydroclimatic variables over the $21^{st}$ century (VIC-CMIP5-RCP8.5), for three future periods: 2010-2039, 2040-2069 and 2070-2099.

### 3.1  Model evaluation

Figure 4a, c, e, g shows monthly time series of observed runoff and VIC-OBS for Rapel, Mataquito, Maule and Itata basins for

1970-2005 period. The observed runoff indicates a high interannual runoff variability for Mataquito, Maule and Itata basins. As discussed before interannual precipitation variability in the region is largely controlled by ENSO (Ruttlant and Fuenzalida, 1991; Falvey and Garreaud, 2009) and consistently, observed runoff tends to increase during El-Niño years (e.g, 1972-1973, 1982-1983 and 1997-1998) and decrease in La-Niña years (e.g., 1973-1974, 1988-1989 and 1998-1999). Overall, VIC-OBS reproduces highly correlated (r=0.8 to 0.9 for annual means) runoff values compared to the observation and it captures ENSO

related interannual runoff variability reasonably well. It is also illustrated that VIC-OBS systematically underestimates low flows and overestimates peak flows in Mataquito and Maule basins (Fig. 4c,e). The similar results were reported by Demaria et al. (2013a).

   Figure 4b, d, f, h presents annual cycle of observed runoff together with VIC-OBS and VIC-CMIP5-HIST for each basin. The observed annual cycle of runoff shows that the four basins are mainly rainfed with 50-70% of annual runoff occuring

during austral winter (JJA). Monthly fractions of runoff for each basin (not shown) indicate that Maule and Mataquito basins receive the highest contribution from melting of snow during austral spring months (SON) ($\sim 36\%$ and $\sim 32\%$, respectively). The VIC-OBS and VIC-CMIP5-HIST runs reproduce the mean annual cycle and give close estimates in Mataquito and Itata




basins (Fig. 4d, h). Whereas, runoff is overestimated for the Maule basin in both VIC-OBS and VIC-CMIP5-HIST (Fig. 4f). Furthermore, a systematic underestimation of low flows during the dry season (November to April) is remarkable in Mataquito and Maule basins.

In addition to runoff comparison, we compare the 2001-2005 mean austral winter (JJA) MODIS-TERRA snow cover (%) with that in the VIC-OBS and VIC-CMIP5-HIST. The VIC model reproduces the observed spatial pattern of mean JJA snow cover reasonably well with high spatial pattern correlations (see Fig. S2b, c, d). In terms of evapotranspiration, we show VIC-CMIP5-HIST comparison with the GLEAM dataset for 1976-2005 period. With respect to GLEAM, VIC-CMIP5-HIST underestimates evapotranspiration and annual cycle comparison indicates that there is a striking underestimation of evapotranspiration in summer for each basin. On the other hand, during the late winter and spring, there is a remarkable agreement and the model reasonably reproduces the peak values. It should be noted that diagnostic evapotranspiration products such as GLEAM have important uncertainties (Mueller et al., 2013). Furthermore, large uncertainties in simulated evapotranspiration by land surface models have been reported in several studies (e.g., Dolman and De Jeu, 2010; Badgley et al., 2015; Garrigues et al., 2015).

### 3.2 Hydroclimatic projections

In this section, changes in runoff, snow water equivalent (SWE), evapotranspiration (ET) and soil moisture (SM) are assessed at annual, seasonal, monthly and daily time scales from the VIC-CMIP5-RCP8.5 simulations. SM is calculated by sum of SM in each three soil layers at 0.1, 0.9 and 1.7 m. Statistically significant differences were computed for each model using the Student's t-test with a 95% confidence level and the grids were hatched accordingly in the spatial map plots where most of the models (>50%) have statistically significant differences. Finally, we analyze daily runoff changes in terms of probability density function (PDF) and return periods in order to assess changes in hydroclimatic extremes.

#### 3.2.1 Changes in climate variables

Figure 5 shows maps of the relative differences in precipitation (top panel) and absolute changes in maximum (middle panel) and minimum temperatures (bottom panel) between VIC-CMIP5-RCP8.5 and VIC-CMIP5-HIST obtained for the three periods of analysis. Towards 2010-2039, precipitation is projected to decrease by up to 10%. In the same period, the simulated temperature increases between $0.5^oC$ (coastal range) to up to $1.5^oC$ over the Andes Mountain, i.e., there is a sharp west-east gradient in the projected temperature increase (Fig. 5b). For the second period (2040-2069) precipitation is projected to decrease between 15-20%, most of the central valley and up to the coast. During this period (mid-century) temperature increases between $1.4-2^oC$ near the coast and central valley and up to $2.5^oC$ in the high Andes. Finally, by the end of the century, most of the domain shows precipitation decreases between 30-35%, somewhat smaller decreases on the mountain tops (25-30%). Temperature increases by the end of the century go from $2.6-3.2^oC$ near the coast, to up to more than $3.6^oC$ over the Andes, again with a very sharp west-east gradient (Fig. 5b).

Comparing the changes in maximum and minimum temperatures (Fig. 5b, c), some differences are apparent. There is a clear west-east temperature increase in maximum temperatures, with larger maximum temperature increases over the Andes





compared to the coast. In minimum temperature, in addition to the west-east gradient the is also a south-north gradient, so that the north-east corner of the domain is the one with the largest increases in minimum temperatures. Maximum temperatures increase more than minimum temperatures especially in 2040-2069 and 2070-2099 periods, therefore increasing the diurnal temperature amplitude.

It is worth mentioning that the bias corrected precipitation changes result in both a higher precipitation decrease compared with the unbiased CMIP5 models, and with a different spatial structure (not shown).

As noted before, the CMIP5 multi-model ensemble mean shows a robust drying in all basins. In order to assess the changes in the probability of having extended droughts due to decreased precipitation, we defined a persistent drought metric as as five or more consecutive years having a deficit equal or larger than 20% with respect to reference period (1976-2005) climatology.

This threshold is similar to the observed precipitation deficit that has occurred during the 2010-2015 period. Fig. 6 shows the persistent drought frequency (events per 100 years) for the reference (1976-2005) and future periods (2010-2039, 2040-2069, and 2070-2099). The CMIP5 data indicates this type of events to become more frequent particularly over the coastal area of CC, and there is a dramatic increase in drought conditions by the end of the century over almost all of the study region (up to 5 events/100years).

**3.2.2   Changes in hydroclimatic regimes**

Figure 7 shows projected annual runoff, ET and SM changes (%) of the ensemble mean VIC-CMIP5-RCP8.5 for the periods of 2010-2039, 2040-2069 and 2070-2079 relative to the VIC-CMIP5-HIST. Runoff is projected to decrease in each successive period with respect to the reference period. The amplitude of the simulated runoff decline is particularly large, of up to 10% in 2010-2039 period and more than 40% by the end of the century. The results are consistent with the overall decrease in

precipitation that contributes to dry conditions in the lowlands and coastal area. ET is projected to have very little or no change in the first period (Fig. 7b). Towards 2040-2069, ET is projected to decrease up to 10% in Mataquito, Maule and Itata basins, and more than 10% in Rapel basin. By the end of the century, although the decrease amplitude is moderate (up to 20%) compared to runoff, ET changes are statistically significant too and a more pronounced decrease is projected to occur in the lowlands and Rapel basin. In some areas of the Andes, ET is projected to increase by the end of the century (albeit not

statistically significant). SM is projected to decrease up to 5% and 10% in mid-century and by the end of century, respectively, in all parts of the basins. Although the decrease in SM is low in mid-century and by the end of century, the changes are statistically significant.

Figure 8 and Table 2 summarize the change in the water balance in each basin in three future periods with respect to the reference period. The ensemble mean VIC-CMIP5-RCP8.5 simulations foresee that precipitation will decrease by 30% in the

basins by the end of the century (see Table 2). Runoff is projected to decrease around 40% and Mataquito and Rapel basins have the highest decreases (40.8% and 40.2%, respectively). The projections indicate that ET declines by 2% to 17% in all basins during the $21^{st}$ century. The highest amount of decrease in ET takes place in Rapel basin by the end of the century (17.3%, see Table 2). The figure illustrates that changes in runoff are more sensitive than those in precipitation with a more pronounced difference by the end of the century indicating the contribution of ET changes. Beside this, as the ratio of annual





evapotranspiration to annual precipitation (ET/P) is expressed as a function of the aridity index (Arora, 2002), it could be stated that an increasing tendency for ET/P, which is more pronounced in the Rapel and Mataquito basins, results in stronger water limited conditions by the end of the century (see Table 2).

An increase in temperature over snowmelt dominated basins should lead to a reduction of SWE and temporal shifts in peak
flow in different regions in the world (e.g., Christensen et al., 2004; Immerzeel et al., 2010; Vicuña et al., 2011; Bozkurt et al., 2015). To further investigate the impact of projected temperature increases on snowpack and runoff, we present the changes in the runoff timing. We adopted center time (CT), that defines the date marking the timing of the center of mass of annual flow (Stewart et al., 2005), to detect the shifts in the runoff timing. Fig. 9 shows fraction of accumulated runoff from the ensemble mean of VIC-CMIP5-HIST and VIC-CMIP5-RCP8.5. The CT of the reference period for the basins is marked as $3^{rd}$
September for the Rapel basin, $11^{th}$ October for the Mataquito basin, $22^{nd}$ August for the Maule basin, and $21^{st}$ July for Itata basin (see Table 2). Projected daily runoff indicates significant temporal shifts to earlier dates in the CT for Rapel, Mataquito and Maule basins for the three future periods (Fig. 9a, b, c). As the increase in temperature is more pronounced by the end of the century, larger shifts to earlier dates in the CT for these basins are projected to occur by the end of the century (∼4 weeks). The most significant shift to earlier dates in the CT takes place in the Mataquito basin with a value of 6 weeks ($11^{th}$ October
to $27^{th}$ August, see Table 2). Accordingly, the Itata basin, which has the lowest snow coverage fraction and is therefore mainly a rainfed basin, has no CT shift (Fig. 9d). Coherently to our finding, Cortés et al. (2011) noted that there is not a significant correlation between temperature (thus snow melting) and CT in the lower basins of the extratropical western Andes Cordillera, while precipitation timing plays an important role, as it determines less snow over these watersheds. Temperature-driven earlier snow-melting also leads to increase in winter runoff in some parts of the Andes (see Fig. S4a).

The shifts in CT towards earlier dates, as well as the decrease in projected runoff, are consistent with SWE changes. The ensemble mean VIC model simulations of SWE depict that the Andes snowpack are projected to be less than half of what they are in the reference period by mid-century (see Fig. S5). Furthermore, relative decreases in mean annual SWE for these basins reveal that largest snowpack deficits take place in the Maule, Mataquito and Itata basins with values of 87%, 86%, 82%, respectively by the end of the century (see Table 2).

In snowmelt-dominated basins, warming and precipitation related ET changes may also have a significant impact on the regional water cycle and seasonality. Although SM integrates the temporal history different hydroclimatic variables and it is somewhat more difficult to analyze than runoff or precipitation (Hamlet et al., 2007), the change patterns of runoff and ET are also related to SM. Therefore, we also look at the absolute seasonal changes of water balance components (precipitation, runoff and ET) as well as SM for Rapel and Itata basins by the end of the century (Fig. 10). Rapel basin has more snowmelt-
dominated regime with a secondary runoff peak in spring, whereas, Itata basin is characterized as more rainfall-dominated basin with more SM content. In general, runoff change ($\Delta R$) follows the changes in precipitation ($\Delta P$) across the year in both basins. The weak ET change ($\Delta ET$) and the associated low response to climate change may be due to an underestimation of ET in the VIC-CMIP5-HIST as discussed in section 3.1. $\Delta R$ matches more closely $\Delta P$ in Itata basin, as it is more rainfed basin (Fig. 10b). On the other hand, in more snowmelt-dominated basin (Rapel), temperature-driven snow-melting towards the
end of the year contributes to seasonal changes. $\Delta ET$ increases (albeit not too marked) at the end of the winter due to earlier





snowmelt and thus, larger water availability. Furthermore, disappearance of snow cover earlier in the year due to snowmelt leads to decrease in surface albedo in both winter and spring seasons (not shown), and hence, increase in surface net radiation. Generally, as the surface net radiation has a very high correlation with ET (e.g., Arora, 2002; Wang et al., 2007; Adam et al., 2009), increased net radiation over the Andes due to reduced surface albedo would lead to a positive feedback on ET during

winter and spring seasons. Towards the end of the year (late spring and summer) $\Delta P$ is the leading driver and therefore ET decreases due to stronger water limited conditions. Therefore, the role of $\Delta ET$ is more important towards the end of the year, probably dampening the precipitation effect in runoff.

In terms of SM changes ($\Delta \theta$), there is a pronounced decrease in SM towards the winter season. The decrease is more striking in Itata basin coherently to $\Delta P$ and $\Delta R$. The $\Delta \theta$ is relatively lower in Rapel basin towards the winter by following the less

$\Delta P$ and $\Delta R$ in the same period. Although $\Delta P$ is very low in the late spring in Rapel basin, $\Delta \theta$ indicates similar amount of decrease in the late spring compared to that in winter. It could be argued that temperature-driven snow-melting, and thus, significant reduction in snowpack and runoff leads to a secondary decrease in SM in Rapel basin towards late spring.

The impact of SM changes on climate is strongest in transitional regions between dry and wet climates (Seneviratne et al., 2010) and several studies have highlighted the important role of SM reduction in altering land-atmosphere feedbacks (e.g.,

Diffenbaugh et al., 2007; Alexander, 2011). Therefore, one can expect that projected dry soil condition in CC could lead to intensification of positive SM-atmosphere feedbacks, and a strengthened future warming-related extremes such as droughts and wildfires.

### 3.2.3 Changes in runoff extremes

In this subsection, we present daily runoff changes in terms of PDFs and return periods in order to illustrate changes in

hydroclimatic extremes. Fig. 11 shows the PDFs of maximum daily runoff for each basin from the ensemble mean of VIC-CMIP5 simulations for the reference period (1976-2005) and future periods. The PDFs of the reference period indicate that the mean and variance of maximum daily runoff increases from north to south (Rapel to Itata basin). For instance, the Rapel basin has the lowest mean value of maximum daily runoff (55.2 mm/day) and has relatively low variance (Fig. 11a). The PDFs of future periods for Rapel and Mataquito basins show that there is an increasing variance especially for the mid-century and end

of the century, which indicates a slight increase in the likelihood of flood events. Maule and Itata basins feature more increase in variance and mean values contributing to much higher likelihood of flood events for each future period (Fig. 11c, d). It is also important to note that the increasing probability of very low maximum runoff in each basin, indicates a higher likelihood of longer period of low flows contributing to the increase in frequency and severity of drought conditions. Much of the increase in the likelihood of higher maximum runoff is mainly caused by changes in the high elevation areas (>1000m), i.e. the Cordillera

(not shown).

Finally we calculated return periods of annual maximum runoff events for each basin from the ensemble mean VIC-CMIP5-HIST and VIC-CMIP5-RCP8.5 simulations for the reference period (1976-2005) and the three future periods (see Fig. S6). We used the open source extRemes package developed by Gilleland and Katz (2015) within R core (R Development Core Team, 2013) for return level calculations. We first applied the block maxima approach (Coles, 2001) by choosing the maximum daily



runoff in each of 30 years for the reference and future periods. By using the maximum daily runoff time series, we then did a fit to the generalized extreme value (GEV) distribution function based on the extreme value theory (EVT). A more detailed description of the method can be found in Gilleland and Katz (2015).

30-year ensemble mean of VIC-CMIP5-HIST simulations for the reference period return level values show that the Maule basin has the greatest maximum runoff with 2, 10 and 50-years return periods (125 mm/day, 147 mm/day and 161 mm/day, respectively) while the Rapel basin has the lowest values (55 mm/day, 69 mm/day and 78 mm/day, respectively, see Table 2). When comparing the projected return period changes with respect to 1976-2005 reference period, there is an upward shift in the maximum runoff return periods, which are more pronounced in the higher recurrence intervals (see Fig. S6). Moreover, the decrease (increase) in return periods (values) of maximum runoff is more remarkable by mid and end of the century, highlighting the likelihood of more frequent flood events. For instance, by mid-century, 10-year maximum runoff values will be greater than the current 50-year maximum runoff values in all basins (see Table 2). Higher likelihood of high-flow magnitude might be related with earlier snowmelt and soil moisture content (Demaria et al., 2013b). Indeed, more snowpack melting in winter time in the Andes leads to increase in SM content of near-surface soil layers (up to 0.1 m, not shown), and hence, can trigger downslope runoff by impeding infiltration. It should also be noted that higher recurrence intervals (e.g., 50-yr) tend to have larger uncertainties especially for the Rapel and Itata basins by mid-century and by the end of the century. For small return periods (e.g., 2-yr), the ensemble mean VIC-CMIP5-RCP8.5 simulations indicate small changes in the maximum runoff values in the future periods, even slight decrease in the magnitude of maximum runoff for each basin by the end of the century.

## 4  Summary and discussion

We assess the impact of projected changes in temperature and precipitation under the RCP8.5 scenario on hydroclimatic regimes and extremes over four Andean basins in central Chile (CC, $\sim 30 - 40^o$S), namely (from north to south) Rapel, Mataquito, Maule and Itata basins. Historical (1960-2005) and projected (2006-2099), daily precipitation and temperatures from 26 CMIP5 climate models are bias corrected and used to drive the VIC model (VIC-CMIP5-HIST and VIC-CMIP5-RCP8.5) to obtain regional hydroclimate projections. Changes in hydroclimatic variables such as runoff, snow water equivalent (SWE), evapotranspiration (ET) and soil moisture (SM) are assessed with respect to the reference period at annual, seasonal, monthly and daily time scales from the VIC model simulations. Our key projection results are:

– CC is projected to become warmer and drier under RCP8.5 scenario. The ensemble mean VIC-CMIP5-RCP8.5 simulations foresee that precipitation will decrease by 30% in the basins by the end of the century, whereas runoff is projected to decrease around 40%. Minimum and maximum temperature are projected to increase up to 3$^o$C and 4$^o$C, respectively, with largest warming over of Andes.

– The projected precipitation decreases increase the probability of having extended droughts, such as the recently experienced mega-drought (2010-2015), to up to 5 events every 100 year.



– Warming is associated with temporal shifts in peak flow and losses of snowpack. Results indicate temporal shifts to earlier dates (3-6 weeks) in the peak timing, except in the Itata basin. The most significant shift to earlier dates in the center time takes place in the Mataquito basin with a value of 6 weeks ($11^{th}$ October to $27^{th}$ August). The Andes snowpack is projected to be less than half of the reference period already by mid-century. Similar results are reported for other semi-arid basins of the Andes in Chile (e.g., Vicuña et al., 2011; Demaria et al., 2013b).

– Changes in the annual cycle of water balance components indicate that runoff change follows the changes in precipitation across the year due to a weak ET change. The role of ET change is more important towards the end of the year and mainly depends on water and SM availability. Reduced surface albedo and its positive feedback onto ET in the Andes also leads to increase in water limited conditions in late spring and summer.

– The probability density functions (PDFs) of annual maximum daily runoff for each basin indicate an increase in the likelihood of daily maximum runoff values, and hence, floods, which is more pronounced for the Maule and Itata basins by mid and end of the century, i.e. for the more rainfed basins of the four basins considered.

– Increased probability of very low maximum runoff in each basin, indicates a higher likelihood of longer period of low flows contributing to the increase in frequency and severity of drought conditions.

– The estimated return periods of annual maximum total runoff events indicate a decrease (increase) in return periods (values). For instance, by mid-century, 10-year maximum runoff values will be greater than the current 50-year maximum runoff values in each basin. Higher likelihood of high-flow magnitude might be related with earlier snowmelt.

Starting from 2010, CC has already been facing a precipitation deficit of approximately 30% together with a clear warming trend in the Andes cordillera in this region, exacerbating the water deficit (CR2, 2015). Beside this, Boisier et al. (2016) showed that natural climate variability (i.e., sea surface temperature variability and associated internal atmospheric variability) plays an important role in explaining the amplitude of the current drought. Our projection results highlight that increasing temperature and decreased precipitation is highly likely to continue and strongly impact the future hydroclimatic conditions of CC and hence to aggravate water and drought stress. For instance, temporal change in peak flows and earlier snowmelt will have implications for water resource management. Barnett et al. (2005) highlighted that without adequate water storage capacity, much of the winter runoff will immediately be lost to the oceans in snowmelt-dominated regions, that will lead to regional water shortages. This alteration of the hydrological cycle in CC will lead to a serious reduction in dry-season water. This is specially true for the three further northern basin of our study region. In addition to excessive water withdrawals, changing hydroclimatic conditions may further exacerbate the consequences of future droughts. For instance, the robust decrease in runoff and SM will not only result in meteorological drought but also in hydrological drought, as well as agricultural drought. The combination of these different drought types will have the potential to negatively affect agricultural activities. Meza et al. (2012) provide an example of irrigation demands of agriculture activities for another basin of CC, finding that water demand from irrigated agriculture tends to increase as a consequence of projected precipitation and temperature changes, which might lead to oversubscription



of water rights. Such water related problems will undoubtedly affect the development of other socio-economic sectors such as tourism and energy.

Changes in hydroclimatic conditions combined with land-cover changes are very likely to have a range of different effects on ecosystems and the mountain cryosphere too. Several studies in snowmelt-dominated regions reported that changes in geological settings such as rock slope stability due to the disappearance of snow cover and downwasting of glaciers, will result in the rise of the likelihood of natural hazards such as landslides, rockfalls and debris flows especially in foothills regions (e.g., Bozkurt and Sen, 2013; Gobiet et al., 2014). Indeed, for instance, higher likelihood of high-flow values due to earlier snowmelt in the Andes is likely to lead to occurrence of natural hazards. In addition to these expected changes, one of the most concerning impacts is the increased number and length of forest fires. CR2 (2015) reported that the area of forest destroyed by fire in central and southern Chile has increased 70% during the 2010-2015 mega-drought. Therefore, combined effects of higher temperature, drier soil conditions and land-cover changes increase the risk of wildfires in CC.

These projected changes could potentially have a major impact on the socio-economic development of this densely populated region in Chile. On one hand, detailed investigations and adaptation efforts are needed to increase resilience and reduce vulnerability, through collaboration and partnerships between various sectors and stakeholders. On the other hand, more research is warranted to clarify dynamics and physical mechanisms as well as regional feedback processes affecting the changes in hdyroclimatic patterns of CC.

*Author contributions.* DB performed the VIC simulations and hydrological analyses and wrote the text. MR carried out the statistical bias correction. JPB carried out the drought calculations and hydrological analyses. JV performed the initial station data analysis. All authors discussed the results and contributed to editing the text.

*Acknowledgements.* We acknowledge the World Climate Research Programme Working Group on Coupled Modelling, which is responsible for CMIP, and we thank the climate modeling groups (listed in Table S1 in the supplementary materials) for producing and making available their model output. For CMIP the U.S. Department of Energy Program for Climate Model Diagnosis and Intercomparison provides coordinating support and led development of software infrastructure in partnership with the Global Organization for Earth System Science Portals. This work was funded by FONDAP-CONICYT 15110009. DB acknowledges support from FONDECYT grant 3150036. MR acknowledges support from NC120066 and CONICYT grant 1131055. JPB acknowledges support from FONDECYT grant 3150492. In particular, we are thankful to Justin Sheffield (Princeton University) and Edwin P. Maurer (Santa Clara University) for providing the VIC model parameter files and gridded meteorological fields.



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



**Table 1.** List of simulations performed in this study

| Simulation Name | Period | Forcing |
|---|---|---|
| VIC-OBS | 1970-2005 | Observation-based gridded dataset (Demaria et al., 2013a) |
| VIC-CMIP5-HIST | 1960-2005 (1976-2005, for comparison) | Bias corrected CMIP5 Historical (26 GCMs) |
| VIC-CMIP5-RCP8.5 | 2006-2099 | Bias corrected CMIP5 RCP8.5 (26 GCMs) |



**Table 2.** Summary of changes (%) in hydroclimatic variables (annual precipitation, runoff, snow water equivalent, evapotranspiration and soil moisture) with respect to reference period. Also included are ratio of annual evapotranspiration to annual precipitation, dates of center of timing and return periods reference and future periods

| | **1976-2005** | | | **2010-2039** | | | **2040-2069** | | | **2070-2099** | | |
|---|---|---|---|---|---|---|---|---|---|---|---|---|
| **Rapel** | | | | | | | | | | | | |
| Precipitation (%) | | | | -7.9 | | | -14.5 | | | -30.4 | | |
| Total Runoff (%) | | | | -10.3 | | | -17.9 | | | -40.2 | | |
| SWE (%) | | | | -28 | | | -49 | | | -73 | | |
| ET (%) | | | | -4 | | | -8.8 | | | -17.3 | | |
| Soil Moisture (%) | | | | -1.7 | | | -3.8 | | | -7 | | |
| ET/P | | 0.43 | | | 0.45 | | | 0.47 | | | 0.52 | |
| CT (date) | | 3-Sep | | | 23-Aug | | | 17-Aug | | | 8-Aug | |
| Ret. Per. (mm/day) | *2yrs* | *10yrs* | *50yrs* | *2yrs* | *10yrs* | *50yrs* | *2yrs* | *10yrs* | *50yrs* | *2yrs* | *10yrs* | *50yrs* |
| | 55 | 69 | 78 | 58 | 79 | 93 | 58 | 99 | 150 | 50 | 75 | 93 |
| **Mataquito** | | | | | | | | | | | | |
| Precipitation (%) | | | | -7.9 | | | -15.8 | | | -30.6 | | |
| Total Runoff (%) | | | | -10.5 | | | -20.8 | | | -40.8 | | |
| SWE (%) | | | | -35 | | | -63 | | | -86 | | |
| ET (%) | | | | -2.8 | | | -6.3 | | | -12.5 | | |
| Soil Moisture (%) | | | | -1.7 | | | -4.1 | | | -7.6 | | |
| ET/P | | 0.36 | | | 0.39 | | | 0.41 | | | 0.46 | |
| CT (date) | | 11-Oct | | | 30-Sep | | | 14-Sep | | | 27-Aug | |
| Ret. Per. (mm/day) | *2yrs* | *10yrs* | *50yrs* | *2yrs* | *10yrs* | *50yrs* | *2yrs* | *10yrs* | *50yrs* | *2yrs* | *10yrs* | *50yrs* |
| | 88 | 106 | 118 | 92 | 118 | 132 | 94 | 127 | 148 | 79 | 116 | 147 |
| **Maule** | | | | | | | | | | | | |
| Precipitation (%) | | | | -6.5 | | | -15.5 | | | -30.1 | | |
| Total Runoff (%) | | | | -8.1 | | | -19.7 | | | -38.8 | | |
| SWE (%) | | | | -35 | | | -63 | | | -87 | | |
| ET (%) | | | | -2.1 | | | -4.8 | | | -8.9 | | |
| Soil Moisture (%) | | | | -1.8 | | | -4.4 | | | -8.1 | | |
| ET/P | | 0.29 | | | 0.31 | | | 0.33 | | | 0.38 | |
| CT (date) | | 22-Aug | | | 8-Aug | | | 3-Aug | | | 30-Jul | |
| Ret. Per. (mm/day) | *2yrs* | *10yrs* | *50yrs* | *2yrs* | *10yrs* | *50yrs* | *2yrs* | *10yrs* | *50yrs* | *2yrs* | *10yrs* | *50yrs* |
| | 125 | 147 | 161 | 141 | 188 | 225 | 137 | 178 | 203 | 118 | 163 | 199 |
| **Itata** | | | | | | | | | | | | |
| Precipitation (%) | | | | -4.9 | | | -14.8 | | | -30.1 | | |
| Total Runoff (%) | | | | -5.6 | | | -18.1 | | | -37.8 | | |
| SWE (%) | | | | -23 | | | -57 | | | -82 | | |
| ET (%) | | | | -3 | | | -7.4 | | | -13.2 | | |
| Soil Moisture (%) | | | | -1.8 | | | -4.7 | | | -8.6 | | |
| ET/P | | 0.31 | | | 0.32 | | | 0.34 | | | 0.39 | |
| CT (date) | | 21-Jul | | | 17-Jul | | | 20-Jul | | | 25-Jul | |
| Ret. Per. (mm/day) | *2yrs* | *10yrs* | *50yrs* | *2yrs* | *10yrs* | *50yrs* | *2yrs* | *10yrs* | *50yrs* | *2yrs* | *10yrs* | *50yrs* |
| | 120 | 136 | 158 | 143 | 179 | 265 | 136 | 216 | 303 | 115 | 161 | 312 |





**Figure 1.** Central Chile basins on a digital elevation map derived from global hydrological data and maps based on shuttle elevation derivatives at multiple scales (HydroSHEDS) (Lehner et al., 2008). Circles indicate the location of the four stream gauges used in VIC model runoff comparison. Note that gauges located in Mataquito, Maule and Itata basins represent the whole catchment outlet. Due to lack of stream gauge data at the outlet point of Rapel basin, one of the sub-catchment outlet gauges is used for this basin. Also shown are 0.25x0.25 degree resolution VIC grids.





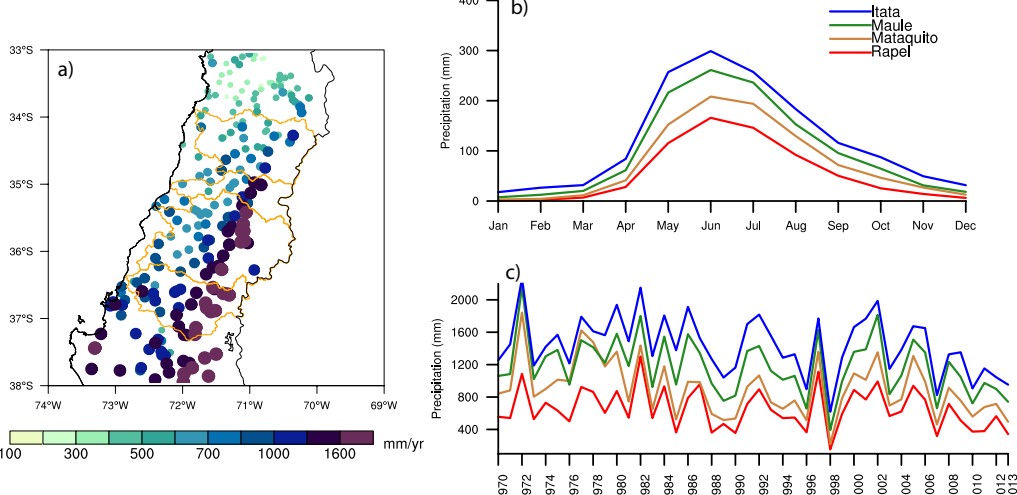

**Figure 2.** (a) 1970-2013 mean annual observed precipitation (mm/yr), (b) mean annual cycle of precipitation (mm), and (c) interannual variation precipitation for each basin.





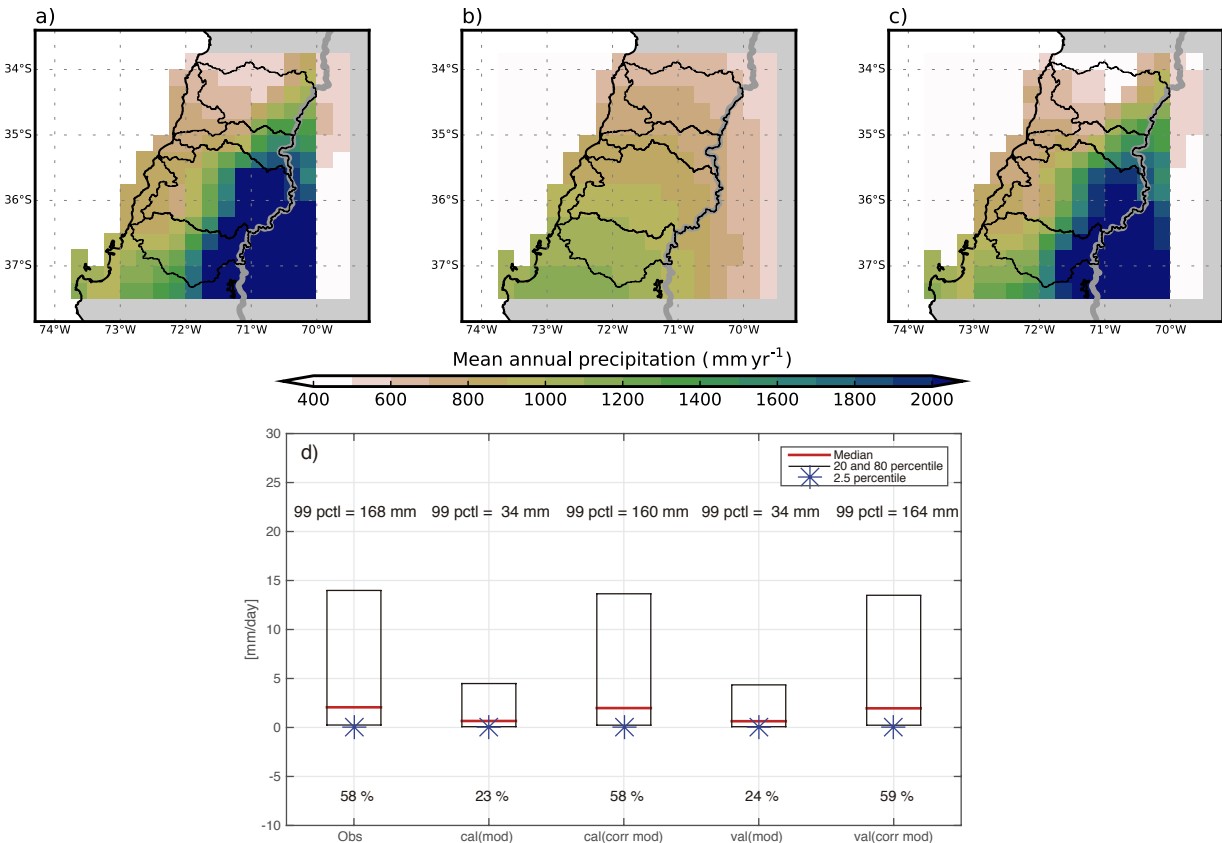

**Figure 3.** (a) Annual mean total precipitation (1976-2005, mm/yr) for gridded observation, (b) CMIP5 ensemble mean uncorrected, and (c) CMIP5 ensemble mean corrected. (d) Also shown is distribution of daily precipitation from observations (obs, left), CMIP5 models for calibration period (1960-1980, cal), CMIP5 models calibration period biased corrected -cal(corr mod), CMIP5 models for validation period (1986-2005, val), and CMIP5 models for validation period biases corrected -val(corr mod). Including percentage of dry days, box of median end percentiles, and the 99th percentile.




**Figure 4.** (a), (c), (e), (g) Observed long-term (1970-2005) monthly runoff (black) and the VIC model simulation result forced with gridded observation (blue) for Rapel, Mataquito, Maule and Itata basins from top to bottom, respectively. Correlation coefficients and root-mean-square errors (RMSE) are calculated by using the annual means. (b), (d), (f), (h) Also shown are annual cycle of observed runoff (black) the VIC model simulation results forced with gridded observation (blue) and CMIP5 models (red for multimodel ensemble mean, gray shading for range of individual models) for Rapel, Mataquito, Maule and Itata basins from top to bottom, respectively.





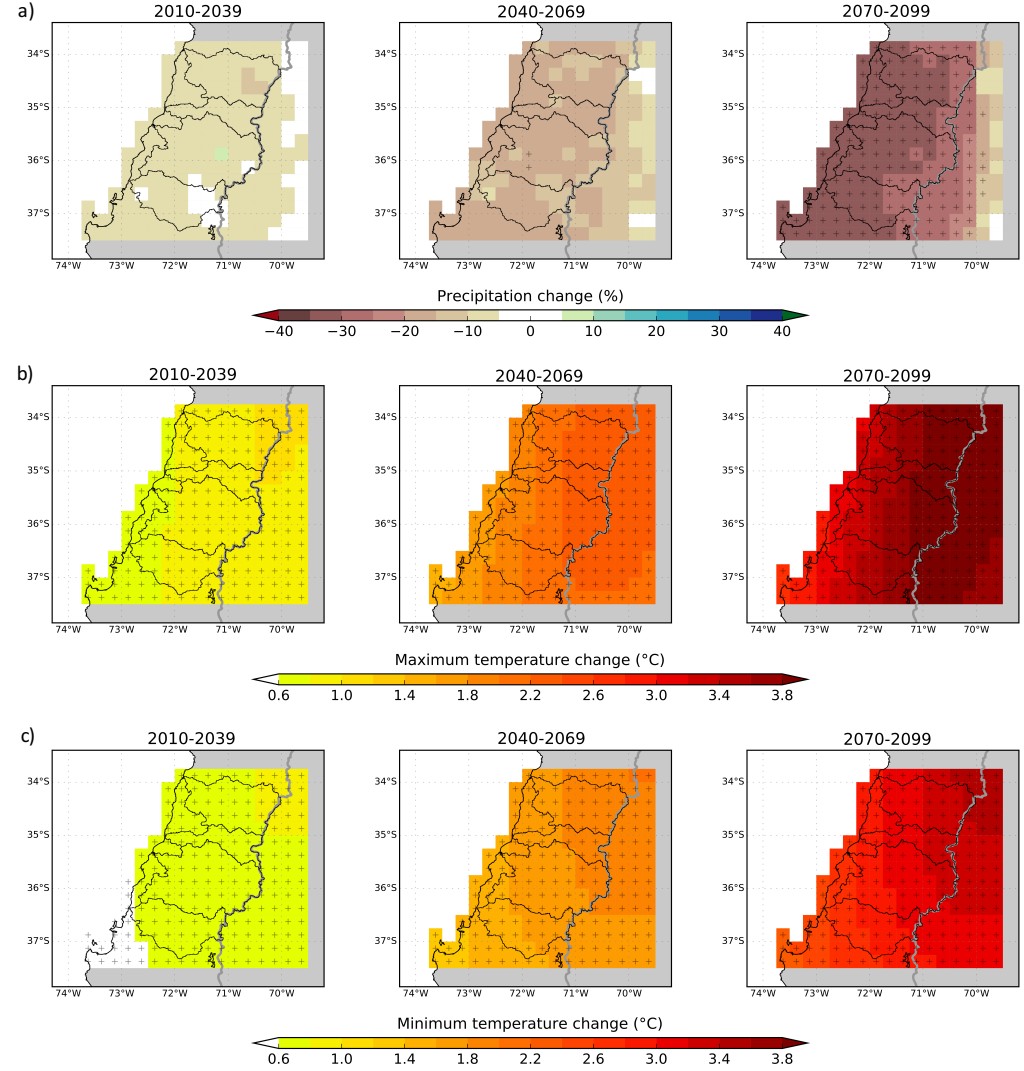

**Figure 5.** (a) CMIP5 ensemble mean bias corrected annual mean precipitation differences (%) for 2010-2039 (left panels), 2040-2069 (middle panels), and 2070-2099 (right panels) periods with respect to 1976-2005 reference period. Also shown are (b) and (c) annual mean maximum and minimum temperature differences ($^o$C), respectively for the same periods with respect to 1976-0025 reference period. Markers indicate where most of the models (>50%) have statistically significant differences at 95% confident level based on Student's t-test.





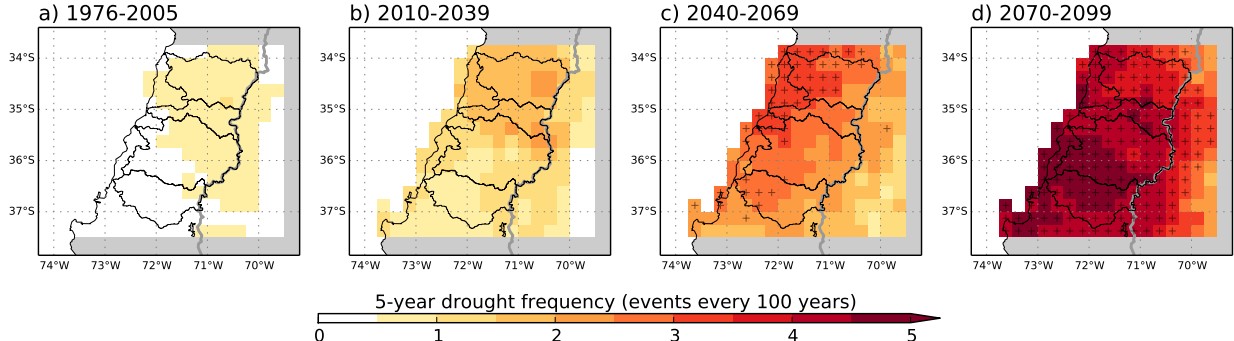

**Figure 6.** (a) Drought events number (every 100 years) for reference period (1976-2005) and future periods (b) 2010-2039 (c) 2040-2069, and (d) 2070-2099. The markers indicate statistically significant changes at 99% confident level based on binomial distribution. Note that drought is defined as 5 or more consecutive years having a deficit equal or larger than 20% with respect to reference period (1976-2005) climatology.



**Figure 7.** (a) Annual runoff, (b) evapotranspiration and (c) soil moisture changes (%) for 2010-2039, 2040-2069, and 2070-2099 periods with respect to 1976-2005 reference period. Markers indicate where most of the models (>50%) have statistically significant differences at 95% confident level based on Student's t-test.




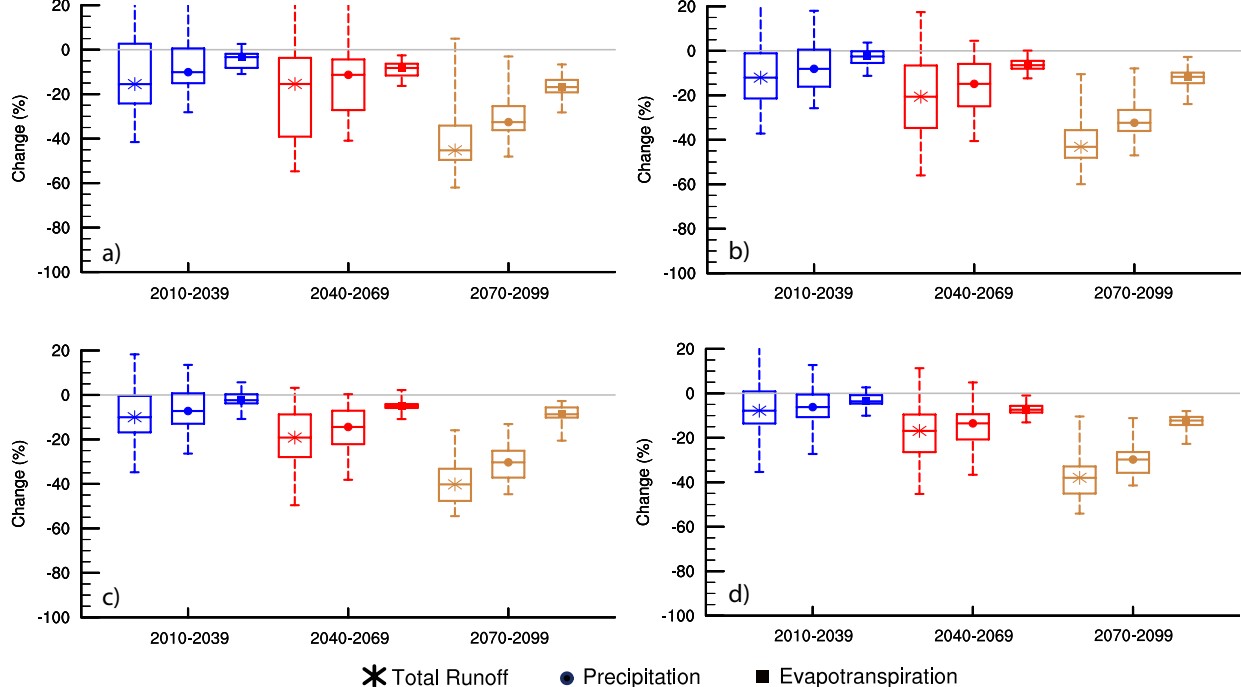

**Figure 8.** (a) Box plots of annual runoff (asterisk), precipitation (dot) and evapotranspiration (square) changes (%) from the ensemble mean of the VIC model simulations forced with CMIP5 models for the periods of 2010-2039 (blue), 2040-2069 (red) and 2070-2099 (brown) relative to 1976-2005 reference period for Rapel, (b) Mataquito, (c) Maule, and (d) Itata basins. The median is represented by the bar across the box and the box-plot whiskers represent the maximum and minimum values. The box represents the 25th and 75th percentiles.





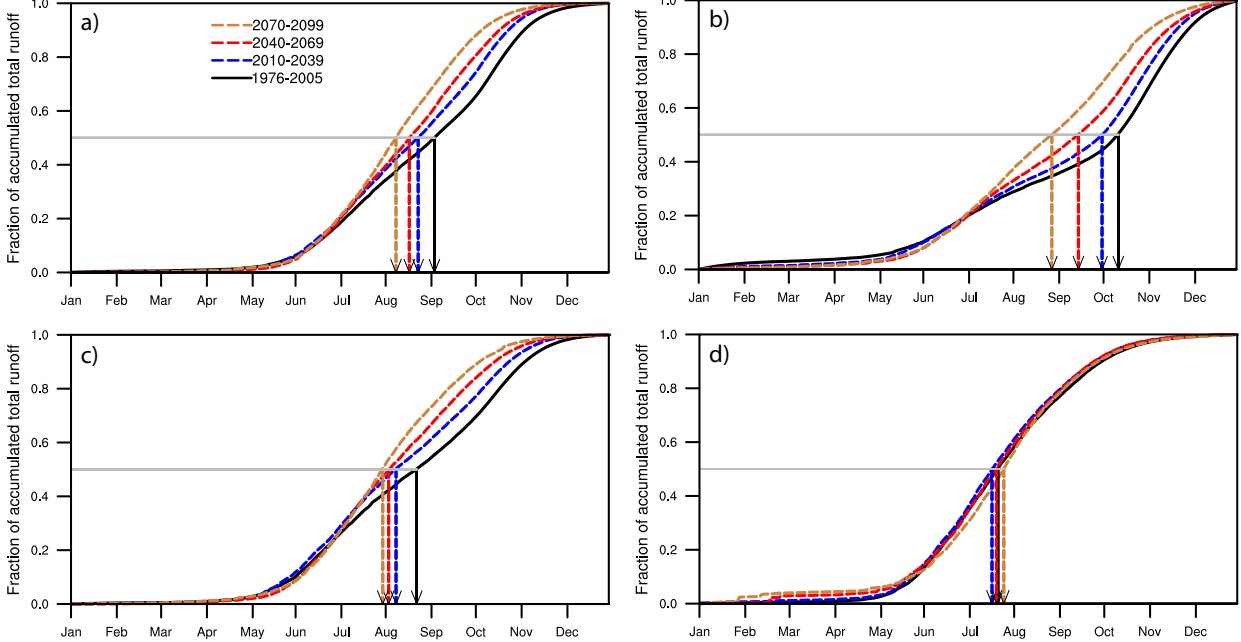

**Figure 9.** (a) Fraction of accumulated runoff from the ensemble mean of the VIC model simulations forced with CMIP5 models for the reference period (continuous line) and future period (dashed lines) for Rapel, (b) Mataquito, (c) Maule, and (d) Itata basins.




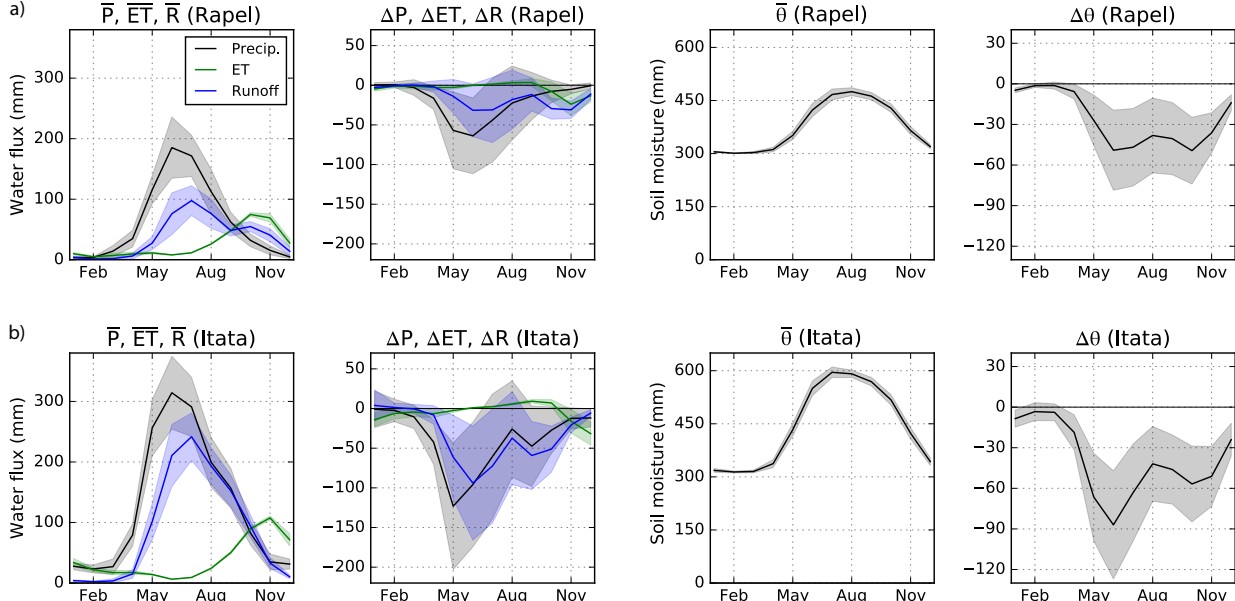

**Figure 10.** (a) Long-term mean annual cycle (mm) of water balance components (precipitation, evapotranspiration and runoff) and their absolute changes (mm) for Rapel basin and (b) Itata basin by the end of the century (2070-2099). Also shown are long-term mean annual cycle (mm) of soil moisture and its absolute change (mm) for the same basins by the end of the century. Shading shows the range of individual models.





**Figure 11.** (a) Probability density functions of daily maximum runoff (mm) from the ensemble mean of VIC model simulations forced with CMIP5 models for the periods of 1976-2005 (black), 2010-2039 (blue), 2040-2069 (red) and 2070-2099 (brown) for Rapel, (b) Mataquito, (c) Maule, and (d) Itata basins.