# Peer review of "Climate change impacts on hydroclimatic regimes and extremes over Andean basins in central Chile"

_Hydrology and Earth System Sciences, 2016_

## Short Comment (SC1) · 6 Feb 2017

First, I would like to clarify that this is not a formal review of this manuscript, but rather a short comment from a reader. Bozkurt et al. report the uncertainty associated with GCM choice on projected hydrologic changes in Central Chile, with emphasis on precipitation, ET, runoff, and soil moisture. The topic is extremely relevant (Murphy et al. 2004; Vano et al. 2015), and the manuscript contains valuable information for the hydrology community. In my opinion, the manuscript would strongly benefit from more context on hydrologic model performance over the baseline period (1976-2005). In particular, the results from Figure 4 suggest that hydrologic model calibration results (from DeMaria et al.) are not satisfactory. In view of this, I have three specific suggestions:

[Figure]

1. The metrics currently included in Figure 4 are based on mean annual runoff, and the calibration results reported by the previous studies by DeMaria et al. are based on monthly flows. In my opinion, it would be very informative to see evaluation results based on daily time steps, especially considering the analyses of extremes (Figure 11). If the authors have daily observations and simulations of runoff, something like the Kling-Gupta Efficiency (KGE; Gupta et al. 2009) would help with this purpose. More importantly, additional metrics based on signature measures of hydrologic behavior (e.g., Yilmaz et al. 2008; Hrachowitz et al. 2014) would provide information on how VIC is doing on simulating high/low flow volumes, basin flashiness, etc.

2. The calibration of soil parameters in VIC bring the risk of creating artificial changes in basin-averaged water storage. Did the authors check that P~Q + ET over the historical baseline period? If the previous is not true, the model very likely produced increase/decrease of water stored in the soil column, and this could amplify projected monthly changes in Figure 10.

3. Overall, hydrologic modeling decisions such as model structure and parameter values may have large implications on projected climate change impacts (e.g., Wilby 2005; Jiang et al. 2007; Bae et al. 2011; Najafi et al. 2011; Surfleet et al. 2012; Surfleet and Tullos 2013; Vano et al. 2012; Mendoza et al. 2016; Mizukami et al. 2016). Moreover, VIC response to snow parameters is quite sensitive (e.g., Elsner et al. 2014; Mendoza et al. 2015). A discussion on these sources of uncertainties – especially on the VIC parameters included in the calibration process – would help to provide context for model performance and hydrologic change results reported in the paper. The authors could also look at the work by Robert Wilby (e.g., Wilby and Harris 2006; Wilby and Dessai 2010) for further discussion on additional sources of uncertainty.

References

Bae, D.-H., I.-W. Jung, and D. P. Lettenmaier, 2011: Hydrologic uncertainties in climate change from IPCC AR4 GCM simulations of the Chungju Basin, Korea. J. Hydrol., 401,

90–105, doi:10.1016/j.jhydrol.2011.02.012.

Elsner, M. M., S. Gangopadhyay, T. Pruitt, L. D. Brekke, N. Mizukami, and M. P. Clark, 2014: How Does the Choice of Distributed Meteorological Data Affect Hydrologic Model Calibration and Streamflow Simulations? J. Hydrometeorol., 15, 1384–1403, doi:10.1175/JHM-D-13-083.1.

Gupta, H. V., H. Kling, K. K. Yilmaz, and G. F. Martinez, 2009: Decomposition of the mean squared error and NSE performance criteria: Implications for improving hydrological modelling. J. Hydrol., 377, 80–91, doi:10.1016/j.jhydrol.2009.08.003.

Hrachowitz, M., and Coauthors, 2014: Process consistency in models: The importance of system signatures, expert knowledge, and process complexity. Water Resour. Res., 50, 7445–7469, doi:10.1002/2014WR015484.

Jiang, T., Y. D. Chen, C. Xu, X. Chen, X. Chen, and V. P. Singh, 2007: Comparison of hydrological impacts of climate change simulated by six hydrological models in the Dongjiang Basin, South China. J. Hydrol., 336, 316–333, doi:10.1016/j.jhydrol.2007.01.010.

Mendoza, P. A., and Coauthors, 2015: Effects of hydrologic model choice and calibration on the portrayal of climate change impacts. J. Hydrometeorol., 16, 762–780, doi:10.1175/JHM-D-14-0104.1.

Mendoza, P. A., M. P. Clark, N. Mizukami, E. D. Gutmann, J. R. Arnold, L. D. Brekke, and B. Rajagopalan, 2016: How do hydrologic modeling decisions affect the portrayal of climate change impacts? Hydrol. Process., 30, 1071–1095, doi:10.1002/hyp.10684.

Mizukami, N., and Coauthors, 2016: Implications of the Methodological Choices for Hydrologic Portrayals of Climate Change over the Contiguous United States: Statistically Downscaled Forcing Data and Hydrologic Models. J. Hydrometeorol., 17, 73–98, doi:10.1175/JHM-D-14-0187.1. http://journals.ametsoc.org/doi/abs/10.1175/JHM-D-14-0187.1.

Murphy, J., D. Sexton, D. Barnett, G. Jones, M. Webb, M. Collins, and D. Stainforth, 2004: Quantification of modelling uncertainties in a large ensemble of climate change simulations. Nature, 430, 768–772, doi:10.1038/nature02770.1.

Najafi, M. R., H. Moradkhani, and I. W. Jung, 2011: Assessing the uncertainties of hydrologic model selection in climate change impact studies. Hydrol. Process., 25, 2814–2826, doi:10.1002/hyp.8043.

Surfleet, C. G., and D. Tullos, 2013: Uncertainty in hydrologic modelling for estimating hydrologic response due to climate change (Santiam River, Oregon). Hydrol. Process., 27, 3560–3576, doi:10.1002/hyp.9485.

Surfleet, C. G., D. Tullos, H. Chang, and I.-W. Jung, 2012: Selection of hydrologic modeling approaches for climate change assessment: A comparison of model scale and structures. J. Hydrol., 464–465, 233–248, doi:10.1016/j.jhydrol.2012.07.012.

Vano, J. A., T. Das, and D. P. Lettenmaier, 2012: Hydrologic Sensitivities of Colorado River Runoff to Changes in Precipitation and Temperature. J. Hydrometeorol., 13, 932–949, doi:10.1175/JHM-D-11-069.1.

Vano, J.A., J. B. Kim, D. E. Rupp, and P. W. Mote, 2015: Selecting climate change scenarios using impact-relevant sensitivities. Geophys. Res. Lett., 42, 5516–5525, doi:10.1002/2015GL063208.

Wilby, R. L., 2005: Uncertainty in water resource model parameters used for climate change impact assessment. Hydrol. Process., 19, 3201–3219, doi:10.1002/hyp.5819.

Wilby, R. L., and I. Harris, 2006: A framework for assessing uncertainties in climate change impacts: Low-flow scenarios for the River Thames, UK. Water Resour. Res., 42, W02419, doi:10.1029/2005WR004065.

Wilby, R. L., and S. Dessai, 2010: Robust adaptation to climate change. Weather, 65, 180–185, doi:10.1002/wea.543.

Yilmaz, K. K., H. V. Gupta, and T. Wagener, 2008: A process-based diagnostic approach to model evaluation: Application to the NWS distributed hydrologic model. Water Resour. Res., 44, W09417, doi:10.1029/2007WR006716.
* * *

---

## Referee Comment (RC1) · Anonymous Referee #1 · 8 Feb 2017

The manuscript is devoted to a study of climate change impacts on hydroclimatic regime in the Andean medium-scale basins. This regional study is important, but the applied methods have several serious deficiencies, and therefore the obtained results are questionable.

The study should be re-done, by substantially improving the model performance, and only after that applying climate scenarios for impact assessment.

(The second part of the manuscript describing simulated climate change impacts was not checked yet.)

1. The model evaluation and its satisfactory performance are prerequisites for impact

[Figure]

**HESSD**

assessment. However, it was not done properly in this study. The authors refer to another study, by Demaria et al. (2013), where "reasonable agreement of VIC model" for this region was shown (from the manuscript is not clear - was it done for one basin, or for all 4 basins). However, the paper by Demaria et al. used 12 GCMs from CMIP3 "to evaluate climate-attributed changes in the hydrology of the Mataquito river basin in central Chile, South America", and not all four catchments. That is why this reference is not fully eligible. Besides, it is not clear, whether the authors used the model setup and parametrization from the former study, or not.

2. The demonstrated results of model evaluation (Figs. 4, S2, S3) are not convincing: - discharge with the monthly time step: results not clear from the graphs, please add criteria of fit, e.g. NSE, PBIAS, RMSE, - seasonal dynamics (Fig. 4 and Fig. S3): runoff is notably overestimated for two catchments of four, ET is underestimated in all four, criteria of fit are missing. For the seasonal dynamics please use Pierson coef. of correlation and bias in standard deviation (see Gudmundsson et al., (2012). Such model evaluation in a regional-scale study with quite weak performance cannot serve as a basis for climate impact assessment, and should be improved.

3. The accepted spatial resolution of 0.25 degrees for catchments from 6.300 to 21.100 km2 seems to be too rough for setting up the model and checking its performance. Besides, it is not clear from the manuscript, whether the sub-grid parametrization was done and how. Such a rough spatial resolution could be also a reason for poor model performance. It is recommended to apply a finer disaggregation scheme for the model calibration (despite that the GCM data for further application is available at 0.25 degrees resolution).

4. The ignored river routing could be also a reason for poor model performance. It is recommended to add it.

5. Description of how the bias-correction of GCMs was done is missing.

6. Language should be checked by/with a native speaker.

---

## Referee Comment (RC2) · Anonymous Referee #2 · 19 Feb 2017

Bozkurt et al. provide new insights into climate change impacts on hydroclimatic regimes and extremes in Andean basins of central Chile. The authors used modeled data to obtain their results and draw their conclusions. In general, I found the manuscript by Bozkurt et al. important since Chile is vulnerable to climate-change-driven impacts on the regional water resources. However, while reading the manuscript, major concerns arise. Most, but not all, of my concerns are related to the preferred model and its applicability for the presented case.

1. The authors showed projections of substantially decreasing discharge by the end of the century. The authors did exclusively concentrate on climate change effects on the regional hydrology, which was the main goal of this study of course. However, I

am convinced that land-use change plays also a major role in this specific area. Did the authors also account for such effects? Land-use change effects often outpace the effect of climate change. Assuming increasing temperature (up to 3.6° C; p. 7 line: 30) and declining precipitation (-30-40%), one might expect changes in the potential vegetation and, thus, the potential land-uses. Against the background of this strong climate shift, I was stunned when reading that ET did not change so much. Can the authors provide any number on the relative contributions of both changes?

2. The authors chose decadal periods to analyze climate change effects, which is absolutely fine. If the aim of the study were to quantify changes in hydroclimatic regimes however, a trend analysis would be probably a better approach. To me, the chosen decadal time scale sounds a bit arbitrary and the authors should – at least – explain better why they chose that explicit temporal step.

3. The authors used the VIC model, though they do not explain why this model suits best for their study. The authors only state that Demaria et al (2013) successfully applied the VIC model to the Mataquito basin. I am not convinced that the VIC model is thus necessarily the best choice for all four studied basins.

4. I this context, I also miss information on the parameterizations for the VIC model runs in all four catchments (p. 5, lines 20 ff: "Parameterization and calibration within VIC is performed primarily through adjustments of infiltration parameters, soil layer thickness and baseflow parameters"... and (lines 25-27) that "vegetation and soil hydraulic properties are based on values from Sheffield et al. (2006). Infiltration parameters, soil layer thickness and baseflow parameters are based on values from Demaria et al. (2013a), which calibrated and validated the model in the study area"). Demaria et al (2013a) only calibrated and validated the model for the Mataquito basin? To me it sounds like that the authors applied the same parameter set to all basins, which is not necessarily given. Regarding the mentioned strong gradients, e.g. in topography and climate, I doubt that such a simplification is justified.

5. The authors assume average soil thicknesses > of 1.7m, as indicated (p 7, line 17). Is this average soil depth realistic? From my own experience I can state that soil thickness varies a lot across Southern and Central Chile. Casanova et al. (2013) (The Soils of Chile) provide a valuable overview of Chilean soils and could be used at least as a reference to compare their soil depth assumptions. Did the authors somehow validate the soil thicknesses used in the model?

6. The authors state that ET is hard to estimate which is definitely true. However, arguing that (p. 7, lines10ff) "diagnostic evapotranspiration products such as GLEAM have important uncertainties" and "large uncertainties in simulated evapotranspiration by land surface models have been reported in several studies" is not a valid argument. If the chosen method has (intrinsic?) problems to realistically simulate ET, the authors should ask if this specific approach is appropriate.

7. I am not totally convinced that the spatial resolution suits well the spatial extent of the basins presented here. A grid of 0.25° seems to be a bit rough. Assuming cells of 0.25°*0.25° yields roughly 750 km2. The authors estimate the basins to 13700, 6300, and 11500 km2. This yields few cells only even for the largest basin. At the same time the authors emphasize strong topographic gradients (e.g., p.3, lines 8-9), which challenges the rough spatial discretization. I suggest considering a finer spatial resolution.

8. The authors state (p. 6, lines 13-14:) that "based on a detailed streamflow validation, Demaria et al. (2013a) illustrated a reasonable agreement between the VIC model simulations and observed fields." The authors point out that the model predictions and the observed values are highly correlated (r=0.8 to 0.9). Based on what correlation is that number derived? The chose of a suitable measure is not trivial in this area that shows strong hydroclimatic seasonality.

9. I am not completely convinced about the outcomes of the return period calculation. Floods are stochastic and, thus, a resampling method (such as bootstrapping) could

be performed to account for uncertainties. The uncertainties are particularly important for the maximum flows. Slight changes here may cause huge differences. I suggest including uncertainty analysis here.

10. The discussion section is rather poor and does more or less concentrate on the consequences of climate change. I am inclined to say that the presented discussion is rather a conclusion section – if condensed. The discussion section should evaluate the different processes that may explain the modeling results, too. The authors mention glacial retreat or loss of snow cover, which in turn may affect soil moisture etc. However, a discussion on these (among others) is missing.

11. The authors present data on decreasing ET for the modeled periods? Is this purely due to the declining precipitation? I (naively?) expect ET to increase since the temperature increase is substantial over the simulated periods. I miss a discussion here.

12. The section 3.2.2 on changes in hydroclimatic changes mixes results and (a relatively superficial) discussion. I suggest separating both, which facilitates the reader in following the arguments provided by the authors. In general, I think the results and discussion sections should be better separated in order to provide a better evaluating of the findings.

Minor:

1. The section 3.2 "Hydroclimatic projections" belongs rather to the method section and not to the results. 2. I am a bit confused about the rates in temperature and precipitation change over the modeled scenario periods. All reported changes (%) are respective to the reference period or do the rates account for the changes from one modeled period to the next one? 3. P. 10, lines 28-29: In one sentence "much" and "mainly". I suggest rewording this sentence. 4. The authors sometimes flip between using passive and active wording, e.g. p. 11, lin 19 and 24: "We assess..." and "...is assessed". I suggest staying consistent and using one way of spelling.

---

## Author Comment (AC1) · 4 Apr 2017

1. The metrics currently included in Figure 4 are based on mean annual runoff, and the calibration results reported by the previous studies by DeMaria et al. are based on monthly flows. In my opinion, it would be very informative to see evaluation results based on daily time steps, especially considering the analyses of extremes (Figure 11). If the authors have daily observations and simulations of runoff, something like the Kling-Gupta Efficiency (KGE; Gupta et al. 2009) would help with this purpose. More importantly, additional metrics based on signature measures of hydrologic behavior (e.g., Yilmaz et al. 2008; Hrachowitz et al. 2014) would provide information on how VIC is doing on simulating high/low flow volumes, basin flashiness, etc.

[Figure]

Response: We acknowledge your comments in this crucial point. Given the fact that the audience of HESS would be interested in seeing more evaluation statistics rather than referring to Demaria et al. (2013a, b), we have expanded the model evaluation part and added more evaluation statistics such as NSE, KGE, PBIAS at monthly and annual time series (results included in table in separate file). Furthermore, we have included two more stream gauges of Itata and Rapel basins so that we could provide a detailed validation statistic for each basin. Finally, we will use three available daily flow data to evaluate model performance in representing CT and PDF for Mataquito, Maule and Itata basins.

2. The calibration of soil parameters in VIC bring the risk of creating artificial changes in basin-averaged water storage. Did the authors check that P~Q + ET over the historical baseline period? If the previous is not true, the model very likely produced increase/decrease of water stored in the soil column, and this could amplify projected monthly changes in Figure 10.

Response: No, we didn't check this, but we see that the VIC model underestimates ET with respect to GLEAM data in summer season. We agree that there may be some inconsistency in the historical and modeled water balances, largely due to ET. We will include another reference dataset (e.g. MODIS) in addition to GLEAM to make our analysis more robust, and have more certainty on the described model biases.

3. Overall, hydrologic modeling decisions such as model structure and parameter values may have large implications on projected climate change impacts (e.g., Wilby 2005; Jiang et al. 2007; Bae et al. 2011; Najafi et al. 2011; Surfleet et al. 2012; Surfleet and Tullos 2013; Vano et al. 2012; Mendoza et al. 2016; Mizukami et al. 2016). Moreover, VIC response to snow parameters is quite sensitive (e.g., Elsner et al. 2014; Mendoza et al. 2015). A discussion on these sources of uncertainties – especially on the VIC parameters included in the calibration process – would help to provide context for model performance and hydrologic change results reported in the paper. The authors could also look at the work by Robert Wilby (e.g., Wilby and Harris 2006; Wilby and Dessai

2010) for further discussion on additional sources of uncertainty.

Response: Although more information on that is given in DeMaria et al. (2013a), we agree that the discussion section indeed lacks of a discussion on sources of uncertainties. We will improve the discussion part adding more content on the sources of uncertainties.

Please also note the supplement to this comment:
http://www.hydrol-earth-syst-sci-discuss.net/hess-2016-690/hess-2016-690-AC1-supplement.pdf

**Supplement:**

Table S1. Model evaluation statistics for monthly and annual time steps for the period 1970-2005

| Basin | r (monthly) | r (annual) | RSR (monthly) | RSR (annual) | PBIAS (%) | NSE (monthly) | NSE (annual) | KGE (monthly) | KGE (annual) |
|---|---|---|---|---|---|---|---|---|---|
| Rapel* | 0.88 | 0.87 | 0.92 | 0.89 | -5.1 | 0.14 | 0.16 | 0.31 | 0.34 |
| Mataquito | 0.81 | 0.82 | 0.77 | 0.68 | -12.6 | 0.4 | 0.53 | 0.63 | 0.77 |
| Maule | 0.87 | 0.86 | 0.71 | 0.6 | -10.8 | 0.5 | 0.62 | 0.6 | 0.82 |
| Itata | 0.91 | 0.91 | 0.44 | 0.48 | -8.6 | 0.8 | 0.77 | 0.87 | 0.83 |

r= Pearson correlation coefficient

RSR= Ratio of RMSE to the standard deviation of the observations

PBIAS= Percent bias

NSE= Nash-Sutcliffe efficiency

KGE= Kling-Gupta efficiency

* Due to many missing values, we choose 1990-2005 period for Rapel Basin.

---

## Author Comment (AC2) · 4 Apr 2017

General response for reviewers 1 and 2:

We appreciate the thoughtful comments by the reviewers. All of them are taken into consideration in our responses. The reviewers raised two major concerns, namely: i) lack of model performance evaluation and ii) the choice of spatial resolution for VIC model simulations.

We agree with both reviewers in that the model setup and simulation protocol adopted in this study may not be suitable for a comprehensive characterization of the hydrological processes within the four basin evaluated. However, we would like first to recall

that the main goal here is to make use of the VIC capabilities to get future basin-wide projections in key hydroclimatic variables (e.g. ET, SWE) that are consistent with the imposed changes in precipitation and temperature. Yet, given the general lack of this type of assessment in central Chile, we considered as a reasonable starting point to extend the work done by Demaria and colleagues over the Mataquito basin to three other basins of the same region (Rapel, Maule and Itata).

For instance, the spatial resolution of 0.25 degree follows the forcing available at the time of performing our simulations (we adopted the one of Demaria et al. 2013). This grid is indeed too coarse to evaluate processes at the scale of secondary and tertiary watersheds in Chile (∼100 to 1000 km2). Aware of this, our group is working on a higher resolution (5km) meteorological forcing for regional hydroclimate research in Chile. This dataset is still in evaluation and will not be used in the present study. The 25km-resolution of the Demaria et al. forcing, although at the limits regarding the main basins assessed in this study (∼10000 km2), is however useful for the general purposes above mentioned.

Understanding the limitations of our study, notably the resolution and the use of the VIC-parameters calibrated for Mataquito for the whole domain, we have considered the reviewer suggestions and improved the model evaluation by adding more statistics such as NSE, KGE, PBIAS. We have found that the VIC model adequately captures relevant information contained in these evaluation metrics, which gives further confidence in the use of this model for hydroclimate change projections in central Chile. Results of those evaluations are given in a separate file as a table.

We think that with these changes, and clearer statement of the goals of the paper will improve the manuscript. Please, find below our point-by-point response to the reviewers' comments.

1. The model evaluation and its satisfactory performance are prerequisites for impact assessment. However, it was not done properly in this study. The authors refer to

another study, by Demaria et al. (2013), where "reasonable agreement of VIC model" for this region was shown (from the manuscript is not clear - was it done for one basin, or for all 4 basins). However, the paper by Demaria et al. used 12 GCMs from CMIP3 "to evaluate climate-attributed changes in the hydrology of the Mataquito river basin in central Chile, South America", and not all four catchments. That is why this reference is not fully eligible. Besides, it is not clear, whether the authors used the model setup and parametrization from the former study, or not.

Response: We acknowledge your comments in this crucial point. Let us first recall that Demaria et al. (2013a) produced a gridded dataset of observed climate (1948-2008) for four basins (Rapel, Mataquito, Maule and Itata) in central Chile. In that study, the VIC model was calibrated to monthly stream flows for the Mataquito basin, and then they applied the same VIC-calibrated parameters in the four basins, as we also did in this study. Demaría et al (2013a) stated that the the rationale for using the same calibrated parameters for the entire domain is to avoid the possibility of allowing extensive calibration to hide the deficiencies of meteorological forcing fields. Demaria et al. (2013a) then validated the VIC model using three gauge sites (one in Mataquito and two in Maule) and based on the calibration/validation statistics they concluded that the VIC model can realistically capture the flow albeit with some biases for high and low flows. Following the same study, Demaria et al. (2013b) used the same validated VIC model to assess the climate change impacts on the hydrology of the Mataquito basin in central Chile through a comparison between 12-member ensembles of CMIP3 and CMIP5. In our study, we follow a similar approach of Demaria et al. (2013b). We used the same model setup and parameterization and extended hydroclimate change projections in central Chile by including three more basins (Rapel, Maule and Itata) and 26 CMIP5 models.

As we have used the same model configuration and parameterization, we did not consider it fundamental to repeat the validation process. Based on your comments and given the fact that the audience of HESS would be interested in seeing more evaluation statistics rather than referring to Demaria et al. (2013a, b), we have expanded the model evaluation and added more evaluation statistics such as NSE, KGE, PBIAS (results included in table in separate file). Furthermore, we have included two more stream gauges of Itata and Rapel basins so that we could provide a detailed validation statistic for each basin. Finally, we have clarified in the text that we used the same model setup and parameterization of Demaria et al. (2013a).

Demaria, E. M. C., Maurer, E. P., Sheffield, J., Bustos, E., Poblete, D., Vicuñna, S., and Meza, F. (2013a): Using a gridded global dataset to characterize regional hydroclimate in central Chile, J. Hydrometeor., 14, 251–265, doi:10.1175/JHM-D-12-047.1.

Demaria, E. M. C., Maurer, E. P., Thrasner, B., Vicuñna, S., and Meza, F. (2013b): Climate change impacts on an alpine watershed in Chile: Do new model projections change the story?, J. Hydrol., 502, 128–138, doi:10.1016/j.jhydrol.2013.08.027.

2. The demonstrated results of model evaluation (Figs. 4, S2, S3) are not convincing: - discharge with the monthly time step: results not clear from the graphs, please add criteria of fit, e.g. NSE, PBIAS, RMSE, - seasonal dynamics (Fig. 4 and Fig. S3): runoff is notably overestimated for two catchments of four, ET is underestimated in all four, criteria of fit are missing. For the seasonal dynamics please use Pierson coef. of correlation and bias in standard deviation (see Gudmundsson et al., (2012). Such model evaluation in a regional-scale study with quite weak performance cannot serve as a basis for climate impact assessment, and should be improved.

Response: Please see our response to your previous point (1). Furthermore, based on your recommendation of Gudmundsson et al. (2012) and also another study of Moriasi et al. (2007), we created a table of evaluation statistics (results included in table in separate file) including the pearson correlation coefficient (r), ratio of RMSE to the standard deviation of the observations (RSR), percent bias (PBIAS), Nash-Sutcliffe efficiency (NSE) and Kling-Gupta efficiency (KGE). Please note that all these statistics are based on monthly and annual time series. Overall, the VIC model performance

is very good in Itata basin and adequate for Maule and Mataquito basins. These results for Maule and Mataquito basins are similar to those in Demaria et al. (2013a, b). On the other hand, the model simulation for Rapel basin shows a poorer performance albeit with low PBIAS and NSE value greater than 0. The reason for this is most probably related with the catchment characteristics of the Rapel basin as it has the highest amount of snow water equivalent among the four studied basins and corresponds to a snowmelt-dominated basin. As noted by Demaria et al. (2013a), while these evaluation statistics do not demonstrate that the best hydrologic model was developed for each basin, we think that the VIC model generally meets the criteria for satisfactory performance and can be used for climate change impact assessment.

Gudmundsson L, Tallaksen LM, Stahl K, Clark DB, Dumont E, Hagemann S, Bertrand N, Gerten D, Heinke J, Hanasaki N, Voss F, Koirala S (2012): Comparing large-scale hydrological model simulations to observed runoff percentiles in Europe. J Hydrometeorol 13:604–620

Moriasi, D. N., J. G. Arnold, M. W. V. Liew, R. L. Bingner, R. D. Harmel, and T. L. Veith (2007): Model evaluation guidelines for systematic quantification of accuracy in watershed simulations. Trans. ASABE, 50, 885–900.

3. The accepted spatial resolution of 0.25 degrees for catchments from 6.300 to 21.100 km2 seems to be too rough for setting up the model and checking its performance. Besides, it is not clear from the manuscript, whether the sub-grid parameterization was done and how. Such a rough spatial resolution could be also a reason for poor model performance. It is recommended to apply a finer disaggregation scheme for the model calibration (despite that the GCM data for further application is available at 0.25 degree resolution).

Response: As we note in the general response above, our selection for spatial resolution is merely based on the data availability of meteorological forcing fields at the moment of performing these simulations. Although clearly not desirable for sub-basins

analyses, we think the grid used is still useful for the purposes of this study.

4. The ignored river routing could be also a reason for poor model performance. It is recommended to add it.

Response: A central objective of the present work is to analyze the projected changes in hydroclimate in central Chile using the basin averages of the grid-based runoff (not routed to basin outlet). In this regard, we aim to illustrate the change in the water balance components (precipitation, runoff and ET) in each basin. As we noted in the manuscript, the region of interest does not include large-scale basins, therefore the summed runoff field does not differ majorly from the river routing (e.g., Shukla and Wood, 2008). Indeed, evaluation statistics for Mataquito and Maule basins in our study are similar to Demaria et al. (2013a) that used a river routing scheme. Please see our response to your previous points 1 and 2 for the detailed model performance evaluation.

Shukla, S. and Wood, A. W (2008).: Use of a standardized runoff index for characterizing hydrologic drought, Geophys. Res. Lett., 35, L02 405, doi:10.1029/2007GL032487.

5. Description of how the bias-correction of GCMs was done is missing.

Response: Our manuscript indeed lacks of exhaustiveness in this point. A more detailed description of the bias-correction methodology will be included in a resubmitted version.

6. Language should be checked by/with a native speaker.

Response: We will check the manuscript for English edits in order to improve its readability.

Please also note the supplement to this comment:
http://www.hydrol-earth-syst-sci-discuss.net/hess-2016-690/hess-2016-690-AC2-supplement.pdf

[Figure]

[Figure]

**Supplement:**

Table S1. Model evaluation statistics for monthly and annual time steps for the period 1970-2005

| Basin | r (monthly) | r (annual) | RSR (monthly) | RSR (annual) | PBIAS (%) | NSE (monthly) | NSE (annual) | KGE (monthly) | KGE (annual) |
|---|---|---|---|---|---|---|---|---|---|
| Rapel* | 0.88 | 0.87 | 0.92 | 0.89 | -5.1 | 0.14 | 0.16 | 0.31 | 0.34 |
| Mataquito | 0.81 | 0.82 | 0.77 | 0.68 | -12.6 | 0.4 | 0.53 | 0.63 | 0.77 |
| Maule | 0.87 | 0.86 | 0.71 | 0.6 | -10.8 | 0.5 | 0.62 | 0.6 | 0.82 |
| Itata | 0.91 | 0.91 | 0.44 | 0.48 | -8.6 | 0.8 | 0.77 | 0.87 | 0.83 |

r= Pearson correlation coefficient

RSR= Ratio of RMSE to the standard deviation of the observations

PBIAS= Percent bias

NSE= Nash-Sutcliffe efficiency

KGE= Kling-Gupta efficiency

* Due to many missing values, we choose 1990-2005 period for Rapel Basin.

---

## Author Comment (AC3) · 4 Apr 2017

General response for reviewers 1 and 2:

We appreciate the thoughtful comments by the reviewers. All of them are taken into consideration in our responses. The reviewers raised two major concerns, namely: i) lack of model performance evaluation and ii) the choice of spatial resolution for VIC model simulations.

We agree with both reviewers in that the model setup and simulation protocol adopted in this study may not be suitable for a comprehensive characterization of the hydrological processes within the four basin evaluated. However, we would like first to recall

that the main goal here is to make use of the VIC capabilities to get future basin-wide projections in key hydroclimatic variables (e.g. ET, SWE) that are consistent with the imposed changes in precipitation and temperature. Yet, given the general lack of this type of assessment in central Chile, we considered as a reasonable starting point to extend the work done by Demaria and colleagues over the Mataquito basin to three other basins of the same region (Rapel, Maule and Itata).

For instance, the spatial resolution of 0.25 degree follows the forcing available at the time of performing our simulations (we adopted the one of Demaria et al. 2013). This grid is indeed too coarse to evaluate processes at the scale of secondary and tertiary watersheds in Chile ($\sim$100 to 1000 km2). Aware of this, our group is working on a higher resolution (5km) meteorological forcing for regional hydroclimate research in Chile. This dataset is still in evaluation and will not be used in the present study. The 25km-resolution of the Demaria et al. forcing, although at the limits regarding the main basins assessed in this study ($\sim$10000 km2), is however useful for the general purposes above mentioned.

Understanding the limitations of our study, notably the resolution and the use of the VIC-parameters calibrated for Mataquito for the whole domain, we have considered the reviewer suggestions and improved the model evaluation by adding more statistics such as NSE, KGE, PBIAS. We have found that the VIC model adequately captures relevant information contained in these evaluation metrics, which gives further confidence in the use of this model for hydroclimate change projections in central Chile. Results of those evaluations are given in a separate file as a table.

We think that with these changes, and clearer statement of the goals of the paper will improve the manuscript. Please, find below our point-by-point response to the reviewers' comments.

1. The authors showed projections of substantially decreasing discharge by the end of the century. The authors did exclusively concentrate on climate change effects on

the regional hydrology, which was the main goal of this study of course. However, I am convinced that land-use change plays also a major role in this specific area. Did the authors also account for such effects? Land-use change effects often outpace the effect of climate change. Assuming increasing temperature (up to 3.6 oC; p.7 line: 30) and declining precipitation (-30-40%), one might expect changes in the potential vegetation and, thus, the potential land-uses. Against the background of this strong climate shift, I was stunned when reading that ET did not change so much. Can the authors provide any number on the relative contributions of both changes?

Response: The changes in vegetation and land use are not accounted for in these simulations. We agree that the land surface forcing may also modulate the future hydroclimatic regime in this highly anthropized region. However, to face such pertinent question we would need to implement a land-cover/land-use change module in the VIC model and perform new sensitivity runs, which is outside the scope of this study. Yet, this point merits a comment in the revised manuscript, so we will add a discussion on this.

2. The authors chose decadal periods to analyze climate change effects, which is absolutely fine. If the aim of the study were to quantify changes in hydroclimatic regimes however, a trend analysis would be probably a better approach. To me, the chosen decadal time scale sounds a bit arbitrary and the authors should – at least – explain better why they chose that explicit temporal step.

Response: To avoid misinterpretations of change due to annual and decadal-scale variability, we use the standard time frame of 30 years (not decadal) to get climatological regimes across different periods of the 21st century. A trend analysis would be an equally useful choice to assess changes, but we prefer the climatology comparison option to highlight and compare in a simple way near-future, mid-term and long-term projections. We will add a short explanation on this in the revised manuscript.

3. The authors used the VIC model, though they do not explain why this model suits

best for their study. The authors only state that Demaria et al (2013) successfully applied the VIC model to the Mataquito basin. I am not convinced that the VIC model is thus necessarily the best choice for all four studied basins.

Response: This main point is partially addressed in our general response above. We have expanded the model evaluation section and added more evaluation statistics such as NSE, KGE, PBIAS (results included in table in separate file). Furthermore, we have included two more stream gauges of Itata and Rapel basins for the evaluation statistics so that we could provide a detailed validation statistic for each basin. As noted by Demaria et al. (2013a), while these evaluation statistics do not demonstrate that the best hydrologic model was developed for each basin, we think that the VIC model generally meets the criteria for satisfactory performance and can be used for climate change impact assessment.

4. In this context, I also miss information on the parameterizations for the VIC model runs in all four catchments (p. 5, lines 20: "Parameterization and calibration within VIC is performed primarily through adjustments of infiltration parameters, soil layer thickness and baseflow parameters". . . and (lines 25-27) that "vegetation and soil hydraulic properties are based on values from Sheffield et al. (2006). Infiltration parameters, soil layer thickness and baseflow parameters are based on values from Demaria et al. (2013a), which calibrated and validated the model in the study area"). Demaria et al (2013a) only calibrated and validated the model for the Mataquito basin? To me it sounds like that the authors applied the same parameter set to all basins, which is not necessarily given. Regarding the mentioned strong gradients, e.g. in topography and climate, I doubt that such a simplification is justified.

Response: We acknowledge your comments in this crucial point. In the study of Demaria et al. (2013a), the VIC model was calibrated to monthly stream flows for the Mataquito basin, then they applied the same VIC-calibrated parameters from the Mataquito basin to the entire domain. In our study, we follow the same approach of Demaria et al. (2013a) and used the same calibrated parameters from the Mataquito basin

to the entire domain, and validated for each basin. Granted, the same VIC-calibrated parameters from the Mataquito basin do not resolve explicitly the strong gradients in topography and climate, but detailed calibration and parameterization analysis is not an objective of our present work. Our major goal is rather to analyze the projected changes in hydroclimate in central Chile using the basin averages of the grid-based runoff using the same validated VIC model from Demaria et al. (2013a) with exactly the same model setup and parameterization. Demaría et al (2013a) stated that the the rationale for using the same calibrated parameters for the entire domain is to avoid the possibility of allowing extensive calibration to hide the deficiencies of meteorological forcing fields. Up to date, there is no higher resolution of gridded observation dataset than the dataset of Demaria et al. (2013a). Understanding the limitations of our study, we have expanded the model evaluation part and added more evaluation statistics such as NSE, KGE, PBIAS and validated the VIC model for each basin (results included in table in separate file).

5. The authors assume average soil thicknesses > of 1.7m, as indicated (p 7, line 17). Is this average soil depth realistic? From my own experience I can state that soil thickness varies a lot across Southern and Central Chile. Casanova et al. (2013) (The Soils of Chile) provide a valuable overview of Chilean soils and could be used at least as a reference to compare their soil depth assumptions. Did the authors somehow validate the soil thicknesses used in the model?

Response: As we have used the same VIC model configuration and parameterization from Demaria et al. (2013a, b), we have refrained from calibration and parameterization in this study. Therefore, we did not validate the soil thickness nor checked the impacts of different soil depths on the results.

6. The authors state that ET is hard to estimate which is definitely true. However, arguing that (p. 7, lines10ff) "diagnostic evapotranspiration products such as GLEAM have important uncertainties" and "large uncertainties in simulated evapotranspiration by land surface models have been reported in several studies" is not a valid argument.

If the chosen method has (intrinsic?) problems to realistically simulate ET, the authors should ask if this specific approach is appropriate.

Response: These sentences are indeed confusing and will be reformulated. Our point here is that there is no gridded ET dataset to be really considered as "observations-driven". Diagnostic, simulated or satellite-derived ET data may be used as a reference but with cautions, as we try to do in our comparison. We are including other reference dataset (e.g. MODIS) in addition to GLEAM to make our analysis more robust, and have more certainty on the described model biases.

7. I am not totally convinced that the spatial resolution suits well the spatial extent of the basins presented here. A grid of 0.25o seems to be a bit rough. Assuming cells of 0.25o*0.25o yields roughly 750 km2. The authors estimate the basins to 13700, 6300, and 11500 km2. This yields few cells only even for the largest basin. At the same time the authors emphasize strong topographic gradients (e.g., p.3, lines 8-9), which challenges the rough spatial discretization. I suggest considering a finer spatial resolution.

Response: As we note in the general response above, our selection for spatial resolution is merely based on the data availability of meteorological forcing fields at the moment of performing these simulations. Although clearly not desirable for sub-basins analyses, we think the grid used is still useful for the purposes of this study.

8. The authors state (p. 6, lines 13-14:) that "based on a detailed streamflow validation, Demaria et al. (2013a) illustrated a reasonable agreement between the VIC model simulations and observed fields." The authors point out that the model predictions and the observed values are highly correlated (r=0.8 to 0.9). Based on what correlation is that number derived? The chose of a suitable measure is not trivial in this area that shows strong hydroclimatic seasonality.

Response: We expanded the model evaluation part and we created a table of evaluation statistics (results included in table in separate file) including pearson correlation

coefficient (r), ratio of RMSE to the standard deviation of the observations (RSR), percent bias (PBIAS), Nash-Sutcliffe efficiency (NSE) and Kling-Gupta efficiency (KGE). Please note that all these statistics are based on monthly and annual time series. Overall, the VIC model performance is very good in Itata basin and adequate for Maule and Mataquito basins. These results for Maule and Mataquito basins are similar to those in Demaria et al. (2013a, b). On the other hand, the model simulation for Rapel basin shows a poorer performance albeit with low PBIAS and NSE value greater than 0. The reason for this is most probably related with the catchment characteristics of the Rapel basin as it has the highest amount of snow water equivalent among the four studied basins and corresponds to a snowmelt-dominated basin. As noted by Demaria et al. (2013a), while these evaluation statistics do not demonstrate that the best hydrologic model was developed for each basin, we think that the VIC model generally meets the criteria for satisfactory performance and can be used for climate change impact assessment.

9. I am not completely convinced about the outcomes of the return period calculation. Floods are stochastic and, thus, a resampling method (such as bootstrapping) could be performed to account for uncertainties. The uncertainties are particularly important for the maximum flows. Slight changes here may cause huge differences. I suggest including uncertainty analysis here.

Response: Thanks for pointing this out. We agree with your comments. In the revised manuscript, we have applied a 3-day moving window to smooth out anomalously large values and recalculated return periods. Furthermore, the figure provided in Supplementary Materials (Fig. S6) includes 95% confidence interval of the return periods.

10. The discussion section is rather poor and does more or less concentrate on the consequences of climate change. I am inclined to say that the presented discussion is rather a conclusion section – if condensed. The discussion section should evaluate the different processes that may explain the modeling results, too. The authors mention glacial retreat or loss of snow cover, which in turn may affect soil moisture etc. However,

a discussion on these (among others) is missing.

Response: The discussion section indeed lacks of content and we will improve it in the revised version.

11. The authors present data on decreasing ET for the modeled periods? Is this purely due to the declining precipitation? I (naively?) expect ET to increase since the temperature increase is substantial over the simulated periods. I miss a discussion here.

Response: As mentioned by Ohmura and Wild (2002) the direction of evaporation trend is not determined by temperature alone. Particularly, in snowmelt basins a complex set of feedback mechanisms, due to both warming and precipitation related changes, may have a great impact on the water balance components at seasonal scale. Indeed, although mean annual ET change illustrates a weak tendency of decreasing, our analysis on absolute seasonal changes of water balance components (pg. 9, lines 25 to 35, and pg. 10, lines 1 to 17) indicates that there is a slightly increase in ET in winter and late winter months, due to earlier snowmelt. However, we agree that that section 3.2.2 is not very clear and in the context of previous comment (10), we will expand the discussion on the physical processes that control the changes.

Ohmura, A. and Wild, M. (2002): Is the hydrological cycle accelerating? Science 298, 1345–1346.

12. The section 3.2.2 on changes in hydroclimatic changes mixes results and (a relatively superficial) discussion. I suggest separating both, which facilitates the reader in following the arguments provided by the authors. In general, I think the results and discussion sections should be better separated in order to provide a better evaluating of the findings.

Response: As noted in the previous comments (10, 11), we agree that section 3.2.2 is not very clear. Therefore, we will clearly separate the results and discussion with more

contents on the evaluation of the different processes.

Minor: 1. The section 3.2 "Hydroclimatic projections" belongs rather to the method section and not to the results. 2. I am a bit confused about the rates in temperature and precipitation change over the modeled scenario periods. All reported changes (%) are respective to the reference period or do the rates account for the changes from one modeled period to the next one? 3. P. 10, lines 28-29: In one sentence "much" and "mainly". I suggest rewording this sentence. 4. The authors sometimes flip between using passive and active wording, e.g. p. 11, lin 19 and 24: "We assess. . ." and ". . .is assessed". I suggest staying consistent and using one way of spelling.

Response: We will put the corresponding information in 3.2 to the method section. All reported changes (%) are respective to the reference period. We will use one way of spelling.

Thank you for your constructive comments.

Please also note the supplement to this comment:
http://www.hydrol-earth-syst-sci-discuss.net/hess-2016-690/hess-2016-690-AC3-supplement.pdf